# Evolution of the eastward shift in the quasi-stationary minimum of the Antarctic total ozone column

**Asen Grytsai[1], Andrew Klekociuk[2,3], Gennadi Milinevsky[1], Oleksandr Evtushevsky[1], Kane Stone[4,5,*]**

[1]Taras Shevchenko National University of Kyiv, 01601 Kyiv, Ukraine

[2]Antarctica and the Global System Program, Australian Antarctic Division, Kingston, Tasmania 7050, Australia

[3]Antarctic Climate and Ecosystems Cooperative Research Centre, Hobart, Tasmania 7000, Australia

[4]School of Earth Sciences, University of Melbourne, Melbourne, Victoria 3010, Australia

[5]ARC Centre of Excellence for Climate System Science, University of New South Wales, Sydney, New South Wales 2052, Australia

[*]Currently at the Department of Earth, Atmospheric and Planetary Sciences, Massachusetts Institute of Technology, Cambridge, Massachusetts 02139-4307, USA.

Correspondence to: G. P. Milinevsky (genmilinevsky@gmail.com)

**Abstract.** The quasi-stationary pattern of the Antarctic total ozone has changed during the last four decades, showing an eastward shift in the zonal ozone minimum. In this work, the association between the longitudinal shift of the zonal ozone minimum and changes in meteorological fields in austral spring (September–November) for 1979–2014 is analysed using ERA-Interim and NCEP–NCAR reanalyses. Regressive, correlative and anomaly composite analyses are applied to reanalysis data. Patterns of the Southern Annular Mode and quasi-stationary zonal waves 1 and 3 in the meteorological fields show relationships with interannual variability in the longitude of the zonal ozone minimum. On decadal time scales, consistent longitudinal shifts of the zonal ozone minimum and zonal wave 3 pattern in the middle troposphere temperature at the southern mid-latitudes are shown. Attribution runs of the ACCESS-CCM chemistry-climate model suggest that long-term shifts of the zonal ozone minimum are separately contributed by changes in ozone depleting substances and greenhouse gases. As is known, Antarctic ozone depletion in spring is strongly projected on the Southern Annular Mode in summer and impacts summertime surface climate across the Southern Hemisphere. The results of this study suggest that changes in zonal ozone asymmetry accompanying ozone depletion could be associated with regional climate changes in the Southern Hemisphere in spring.

**Keywords:** stratosphere, ozone hole, planetary wave, longitudinal shift, ERA-Interim climatology, ACCESS-CCM chemistry-climate model

## 1 Introduction

The distribution of total column ozone over Antarctica is significantly non-uniform during austral spring, i.e. in September–November (Wirth, 1993; Malanca et al., 2005; Grytsai et al., 2007a; Agosta and Canziani, 2010, 2011). This is particularly the case since the early 1980s due to the presence of ozone depletion associated with the ozone hole (Chubachi, 1984; Farman et al., 1985; Chubachi and Kajiwara, 1986; Stolarski et al., 1986; Solomon, 1999). The total ozone distribution

predominantly characterizes the stratosphere layer, where a sharp ozone maximum is usually observed at altitudes of 15–25 km (Chubachi, 1984; Solomon et al., 2005). Therefore, strong variations in the total ozone distribution are due mainly to stratospheric chemistry and dynamics (Wirth, 1993; Gabriel et al., 2011). The ozone hole is located inside polar stratospheric vortex, which is a cyclonic structure that impedes mixing between high-latitude and mid-latitude air masses

(Brasseur et al., 1997). The polar vortex is under the influence of large-scale planetary waves, which disturb the vortex edge region (Wirth, 1993; Quintanar and Mechoso, 1995) and the vortex location relative to the pole (Waugh and Randel, 1999). The scale of the waves in the zonal direction is typically characterized by zonal wave number, which equals to the ratio of the corresponding great circle circumference at a given latitude to the wave length (Hirota and Hirooka,

1984; Hio and Yoden, 2004). The stationary part of the wave structure in the Southern Hemisphere (SH) spring stratosphere is mainly determined by a planetary wave with zonal number 1 (Hartmann et al., 1984; Quintanar and Mechoso, 1995; Ialongo et al., 2012), i.e. wave-1. The role of planetary waves was especially important in the unusual SH stratospheric warming in 2002 (Varotsos, 2002; Allen et al., 2003; Hoppel et al., 2003). Both wave-1 and wave-2 activity during austral winter and

spring caused strong deceleration and warming of the stratospheric polar vortex, its anomalous splitting and ozone hole breakup in September 2002 (Varotsos, 2002; Nishii and Nakamura, 2004; Newman and Nash, 2005; Peters et al., 2007; Grassi et al., 2008; Peters and Vargin, 2015).

In spring, the ozone distribution in the Antarctic region is asymmetrical with a maximum in the Australian longitudinal sector and a minimum in the Atlantic longitudes (Grytsai et al., 2005;

Agosta and Canziani, 2011). Previous studies have revealed the tendency of the Antarctic polar vortex to exhibit an eastward shift in orientation (Huth and Canziani, 2003), in the ozone minimum location (Grytsai et al., 2005; Malanca et al., 2005; Grytsai et al., 2007a, b; Agosta and Canziani, 2010, 2011; Grytsai, 2011; Hassler et al., 2011) and in the phase of wave-1 in stratospheric temperature (Lin et al., 2010). This eastward shift has been described as possibly connected with a

change in tropospheric stationary waves (Grytsai et al., 2007a), tropospheric jet structure (Hio and Hirota, 2002; Agosta and Canziani, 2011) and its strengthening (Wang et al., 2013), and stratospheric ozone and volcanic aerosol concentration (Lin et al., 2010). The quasi-stationary wave (QSW) activity increases typically in austral spring (Randel, 1988) and its enhancement leads to larger vortex asymmetry, a decrease in ozone hole area and net stratospheric ozone loss. It has been

noted that the decreased (increased) asymmetry in the ozone distribution is associated with the eastward (westward) phase shift of the zonal minimum from both observations at the southern high latitudes (Grytsai et al., 2008; Agosta and Canziani, 2011) and climate model simulations for the northern high latitudes (Gabriel et al., 2007).

It has been revealed that the Antarctic ozone depletion in spring influences the trend in the Southern

Annular Mode (SAM) in summer towards the high-index polarity that leads to a range of significant summertime surface climate changes (Thompson et al., 2011). Recent studies have indicated that a stabilization of the spring ozone depletion has occurred from the mid- or late 1990s (Grytsai, 2011; Salby et al., 2011; Kuttippurath et al., 2013; Dameris and Godin-Beekmann, 2014, Solomon et al., 2016). This stabilization relates to the total area of the ozone hole, minimum total ozone values,

ozone mass deficit and duration of the ozone hole season. Chemistry-climate models have displayed a general minimum in Antarctic ozone during 2000–2005 (Siddaway et al., 2013) and slow ozone recovery in the 21$^{st}$ century (Dameris and Godin-Beekmann, 2014). In turn, ozone recovery is expected to continue to impact the SH surface climate (Thompson et al., 2011). This work is focused on the recent tendencies in the zonal asymmetry of the Antarctic total ozone in austral spring and

their possible relations to the SH atmospheric anomalies.

## 2 Data and methods

In this study, gridded monthly-mean satellite measurements of the total ozone column (TOC) are

used to estimate tendencies in the Antarctic quasi-stationary pattern. We restrict our analysis to the September to November (SON) period (austral spring) when the zonal asymmetry in total column ozone is most pronounced. We use measurements from the Total Ozone Mapping Spectrometer (TOMS) / Nimbus-7 (1979–1992), TOMS / Earth Probe (1996–2005) and Ozone Monitoring Instrument (OMI) aboard the Aura platform (2006–2015) which were obtained from the NASA

Ozone & Air Quality website http://ozoneaq.gsfc.nasa.gov/. A gap between Nimbus-7 and Earth Probe observations in 1993–1995 was filled in by Multi Sensor Reanalysis (MSR) data (http://www.temis.nl; van der A et al., 2010).

The original data were obtained with latitude and longitude resolutions of 1° and 1.25°, respectively. We averaged the data over each SON interval to suppress the effects of long-period

travelling planetary waves (Grytsai et al., 2007a), and selected zonal values from 50°S to 80°S at 5° latitude intervals; the centres of our selected latitude bins are 50.5°S, 55.5°S, … 80.5°S. The zonal asymmetry in the TOC distribution over this latitude range is illustrated by the OMI measurements in October 2014 (Fig. 1a). Thick white contour in Fig. 1a shows the ozone hole boundary, which corresponds to the threshold TOC value of 220 DU (white line on colour scale in Fig. 1a) defined by the World Meteorological Organization (WMO) criterion (WMO, 2014); see also (Newman et al., 2004) for the rationale for choosing the 220 DU value.

Figure 1b shows long-term changes in the zonal TOC asymmetry at 65°S (white circle in Fig. 1a) during SON 1979–2014. In the 2000s (blue curves in Fig. 1b), when ozone loss culminated, zonal TOC minimum is about 100 DU lower than in the pre-ozone hole years (red curves) and is notably displaced to the east. This illustrates a general tendency for the zonal TOC minimum to be located further eastward with decreasing minimum level, as noted previously (Grytsai et al., 2005; Lin et al., 2010; Agosta and Canziani, 2011; Hassler et al., 2011). For the quantitative analysis presented here, the longitudes of the zonal TOC maximum and minimum were determined using a 50-degree window in longitude to separate out the large-scale parts of the disturbances. In this work, long-term tendencies were obtained from polynomial approximation calculated with a least-squares method. The calculations method is described in Supplement 1.

Regression, correlation and composite analyses were used to relate the QSW TOC minimum ($QSW_{min}$) longitude to the meteorological variables. The reliability of the main results is examined comparing the relationships with the two reanalysis data. We use gridded atmospheric variables from the European Centre for Medium-Range Weather Forecasts (ECMWF) reanalysis ERA-Interim (Dee et al., 2011; http://www.ecmwf.int/en/research/climate-reanalysis/era-interim) at 1.5°×1.5° (longitude×latitude) resolution and NCEP–NCAR reanalysis (Kalnay et al., 1996; http://www.esrl.noaa.gov/psd/) at 2.5°×2.5° resolution.

## 3 Results

### 3.1 Longitudinal changes in the QSW structure

First, we compare the long-term changes in total ozone (Fig. 2a) and the longitudinal position of the quasi-stationary ozone minimum (Fig. 2b) at 65°S latitude, which is located in the edge region of the ozone hole and polar stratospheric vortex (Roscoe et al., 2012) and where the largest QSW amplitude is observed (Grytsai et al., 2007a; Ialongo et al., 2012).

Cubic polynomial fits are shown by thick curves in Fig. 2. These fits are included to highlight the long-term variations in each of the time series and are not done on consideration of any particular

underlying physical process. In the early 2000s, the decadal tendencies indicated by the slope of the polynomial fit changed sign in both ozone (from decreasing levels to increasing levels, Fig. 2a) and in ozone minimum longitudes (from eastward shift to westward shift, Fig. 2b). Comparison of polynomials from $k = 2$ to $k = 6$ gives similar opposite tendencies before and after the early 2000s (see Supplement 2, Fig. S1). Note that the quasi-stationary maximum in total ozone at 65°S shows a

relatively small decadal shift in longitude (Fig. 2c, thick curve) compared with the level of interannual variability and changes in the $QSW_{min}$ longitude.

    On comparing Fig. 2a and 2b, there appears to be some consistency in the epoch of inflexion in the tendencies of both the column amount and the maximum eastward longitude shift of the zonal TOC minimum (around 2000). The eastward shift in the QSW structure over Antarctica has been

described previously (Huth and Canziani, 2003; Grytsai et al., 2005, 2007a; Malanca et al., 2005; Agosta and Canziani, 2010, 2011; Lin et al., 2010; Grytsai, 2011; Hassler et al., 2011). Eastward shift speeds of about 15–20°/decade are consistent among various studies (Grytsai et al., 2007a; Lin et al., 2010; Hassler et al., 2011). For the period 1979–2000, the time series in Fig. 2b gives a linear trend of 14.4±12.5°/decade, significant at the 95% level. The westward shift between the early

2000s and early 2010s apparent in Fig. 2b is statistically insignificant and a longer time series is necessary to reliably establish this tendency. Note that Fig. 2b shows large longitude variations during some of the most recent years. Interannual changes in the 2000s and early 2010s covered a wider longitudinal range than in the previous decades. For example, the position of the quasi-stationary minimum was near its extreme western values in 2011 and 2013, whereas it reached the

farthest eastern longitude in 2010.

    Long-term tendencies in the QSW minimum/maximum longitudes at the seven latitudes between 50°S and 80°S are illustrated in Fig. 3. The extreme longitudinal departures for 1979, 2002 and 2015 relative to the cubic polynomial are shown by closed circles in Fig. 2b. A significant eastward shift of the $QSW_{min}$ from 1979 (solid blue curve) to the early 2000s (dotted blue curve for 2002) by

30–60 degrees of longitude is seen. The curve for 2015 in the region of the zonal TOC minimum (black dotted curve in Fig. 3) is located between the curves for 1979 and 2002 and is shifted westward over the whole zone 50–80°S, i.e. in opposite direction compared with most years of the preceding decade. Changes of the QSW maximum longitudinal position are not regular (Grytsai et al., 2007a) and the largest eastward shifts are seen only at 55°S and 60°S, however, they are not

statistically significant due to strong interannual variability (Fig. 2c). The relative stability of the zonal maximum location suggests that higher zonal wave numbers, quasi-stationary wave-2 (QSW2) and wave-3 (QSW3), could be present in the QSW structure, in addition to the dominant quasi-stationary wave-1 (QSW1) (Grytsai et al., 2007a; Agosta and Canziani, 2011).

The two curves in Fig. 4 illustrate the similarity in the interannual variations and decadal changes of the $QSW_{min}$ longitudes and ozone mass deficit in Antarctic spring. The linear correlation between the two variables is positive and $r = 0.49$–$0.57$ for the seven latitude circles between 50°S and 80°S with maximum at 60°S. The correlation coefficient $r$ was calculated for the time series length $N = 35$ (1979–2013, Fig. 4) and the value $r = 0.43$ is significant at the 99% confidence limit based on a two-tailed Student's $t$-test. Hence, an eastward shift of the QSW minimum in the ozone distribution with high probability corresponds to a greater ozone mass deficit (larger ozone loss).

Simultaneous negative deviations are observed in the years of large (1988) and major (2002) stratospheric warmings (vertical lines in Fig. 4). Both anomalous events in the SH stratosphere were associated with enhanced planetary wave activity (Varotsos, 2003a; Allen et al., 2003; Baldwin et al., 2003; Grytsai et al., 2008). As seen from Fig. 4, relatively small ozone mass deficits (high total ozone levels) correspond to the westward shift of the $QSW_{min}$ position. Note that this correspondence is not observed in some recent years; the Antarctic spring in 2010 is characterized by eastern longitude of the $QSW_{min}$ at low ozone mass deficit, and the relationship for 2011 is opposite.

The results of Figs. 2–4 show that the eastward shift in the TOC zonal minimum longitude in the Antarctic region was occurred during 1980s–1990s and was accompanied by rapid and intense ozone loss. This decadal tendency appears to have stopped in the early 2000s and became of possibly reverse sign later in the 2000s and 2010s. Generally, the behavior of the zonal TOC minimum in Fig. 2b and Fig. 3 follows the decadal change in the severity of the ozone hole due to international controls on ozone depleting substances (Salby et al., 2011; Solomon et al., 2016), with increasing depletion of the Antarctic ozone in 1980s and 1990s, its leveling off and possible start of recovery in 2000s–2010s (Siddaway et al., 2013). Significant decadal changes in the SH polar ozone are coupled with the stratospheric thermal regime (e.g., Crook et al., 2008) and, because of the zonal asymmetry in the ozone heating, they impact planetary wave propagation (Albers and Nathan, 2012) and regional climate change in both the troposphere and the stratosphere (Gillet et al., 2009; Waugh et al., 2009). Couplings between changes in the QSW structure in Antarctic total column ozone and in atmospheric variables are analyzed below.

## 3.2 Relationships between the QSW minimum longitude and meteorological parameters

To determine the most reliable mean tendencies, we have created a time series for the $QSW_{min}$ longitudes averaged between 55°S and 70°S (four latitude circles in Fig. 3); this is shown in Fig. 5. Bars in Fig. 5 indicate standard deviations for each year and they are generally relatively small compared with the level of interannual variability in the time series.

We next consider the regression between the time series of Fig. 5 and SON average climatological anomalies of ERA-Interim meteorological variables. We first produce monthly climatological anomalies for each gridded monthly average variable at the native horizontal resolution by subtracting the associated long-term monthly mean (over 1979–2014 for ERA-Interim and 1981–2014 for NCEP–NCAR). We then produce averages of the anomalies in grid boxes of $10° \times 10°$ (latitude×longitude) over the SON months of each year. Finally, we evaluated the regression coefficient (RC) between the time series of $QSW_{min}$ longitude of Fig. 5 and the time series of SON climatological anomalies at each location. Figures 6a and 6b present the global distributions of the regressions for surface pressure (SP) and 2 metre air temperature (T-2m) over 1979–2014, respectively. Dots with numbers from 1 to 5 in Fig. 6a are placed at each grid box centre where the linear correlation coefficient is significant at the 95% confidence limit based on Student's $t$-test (boxes with significant correlations are also diagonally hatched) and where the fraction of variance explained (values in %) is $\geq 25\%$ (see text below).

The RC distributions as in Fig. 6a and 6b, but for RC between the $QSW_{min}$ longitude and ERA-Interim 200-hPa climatological anomalies of zonal wind speed (U200) and vertical pressure wind speed (W200) are shown in Fig. 6c and 6d, respectively. The pressure level of 200 hPa corresponds to the upper troposphere in the tropics and lower stratosphere in the SH extratropics and is usually used to analyze the interaction between the tropics and extratropics (e.g. Mo and Higgins, 1998). The RC distribution for sea surface temperature (SST) is similar for that for T-2m (Fig. 6b) and the RC distribution for meridional wind V200 is similar to that for W200 (Fig. 6d), and are not shown here.

The RC distribution in Fig. 6a shows an annular pattern that is similar to a classic Southern Annular Mode pattern in SH climate variability, with pressure or geopotential height anomalies of opposite sign in the middle and high latitudes (Thompson and Wallace, 2000). Negative (positive) regression coefficients in the high (middle) SH latitudes indicate that the $QSW_{min}$ eastward shift is associated with decreased (increased) surface pressure, i.e. with the SAM deviation towards positive polarity.

A positive polarity of the annular mode is accompanied by strengthening of the subpolar westerlies in the SH troposphere and stratosphere and cooling of polar cap regions (Thompson and Wallace, 2000). The RC maximum around 60°S in Fig. 6c and the RC minimum over the Antarctic continent (poleward of 60°S) in Fig. 6b display similar tendencies in the relationships between the $QSW_{min}$ eastward shift and increase of U200 and decrease of T-2m, respectively.

Zonally asymmetric components of the SH circulation, which are most marked in the austral winter and spring (Mo and Higgins, 1998; Fogt et al., 2012a), are also presented in Fig. 6. Three positive RC anomalies in the SH midlatitudes (grid boxes 1, 2 and 3 in Fig. 6a) demonstrate the presence of

a QSW3 structure. The highest negative RC anomaly between grid boxes 2 and 3 is spatially close to the subpolar negative anomaly at grid box 4 and is possibly combined effect of QSW1 and QSW3 (Mo and Higgins, 1998). A significant negative RC anomaly near West Antarctica (grid box 4 in Fig. 6a with an explained variance of 35%) is spatially coincident with the 'pole of variability' in the Amundsen–Bellingshausen Sea Low (ABSL) region (Fogt et al. 2012b and references therein; Turner et al., 2013; Raphael et al., 2016). The midlatitude QSW3 patterns extended to sub-Antarctic latitudes are seen also in the RC distribution for T-2m and W200 (Fig. 6b and 6d, respectively).

The presence of the QSW3 structure in Fig. 6b introduces regional anomalies in the surface temperature distribution. The patterns suggest that when the $QSW_{min}$ moves to the east, surface temperatures are warmer in the Antarctic Peninsula–Weddell Sea region and in the south-west area of Indian Ocean, and cold anomalies appear in the South Pacific and south-east Australia. Hence, variability in zonal asymmetry in the Antarctic ozone during the spring months, with high probability, is indicative of the SH regional climate variability.

Zonal asymmetry in the SH troposphere circulation is closely coupled with the Pacific–South American (PSA) mode (Mo and Higgins, 1998). The PSA pattern in the RC distribution in Fig. 6 is of insignificant intensity, whereas pronounced meridional wave trains are seen in Indian–Australian sector and Atlantic–South American sector (U200 in Fig. 6c). As follows from the relationships below, combined wave activity over the three ocean basins can contribute to the $QSW_{min}$ longitude variability.'

All patterns of Fig. 6 are reproduced in correlations with the same variables using the NCEP–NCAR reanalysis data (Supplement 3, Fig. S2) confirming the reliability of the results. In general, Fig. 6 and Fig. S2 show that interannual variations of the SON $QSW_{min}$ longitude during 1979–2014 are associated with the zonally symmetric annular mode and zonally asymmetric QSW structures in the SH atmosphere. In Fig. 7 we present anomaly composites (averages) for years of extreme western (lower 20[th] percentiles) and eastern (upper 20[th] percentiles) $QSW_{min}$ longitudes to further investigate the patterns shown in Fig 6. Monthly mean anomalies for September, October and November were calculated by subtraction of the climatological means of 1979–2014 from monthly mean variable value in each grid box as described above concerning Fig. 6. Then monthly mean anomalies were averaged over the SON months.

Anomaly composites for the lower 20[th] percentile of the $QSW_{min}$ longitudes (<–60° longitude, 8 westernmost locations outlined by dashed rectangle in Fig. 5a) are presented in Fig. 7a–7c. Anomaly composites for the higher 80[th] percentile (>–3.8° longitude, 8 easternmost locations outlined by solid rectangle in Fig. 5a) are presented in Fig. 7d–7f. The years for the westernmost longitudes are 1979, 1980, 1981, 1988, 1990, 2002, 2011 and 2013 (left column of Fig. 7) and the

years for the easternmost longitudes are 1985, 1992, 1998, 2001, 2003, 2006, 2008 and 2010 (right column of Fig. 7).

It generally is seen from Fig. 7 that transition from the westernmost longitudes (left column) to the easternmost longitudes (right column) is accompanied by the reversal in the sign of the anomalies. The western (eastern) longitudes correspond to negative (positive) zonal wind anomaly around 60°S in Fig. 7a (7d) and positive (negative) surface pressure poleward of 60°S in Fig. 7b (7e). Opposite anomaly combinations appear in the SH middle latitudes. These changes are consistent with the

regression maps in Fig. 6a and 6c, where eastward phase shift indicates similar relationships with U200 (Fig. 6c) and SP (Fig. 6a) displaying the SAM pattern.

Note that the SP anomaly composites show the QSW3-like structure in the SH midlatitudes, which is more intense and shifted to the east in the case of the easternmost $QSW_{min}$ longitudes than in the case of the westernmost longitudes (Fig. 7e and 7b, respectively). The pattern of Fig. 7e means that

the regional surface pressure becomes higher in the eastward shifted elements of the QSW3 structure, i.e. deviates to anticyclonic regime in the case of the easternmost $QSW_{min}$ longitudes. The surface pressure growth between the QSW3 anomalies in Fig. 7b and 7e equates to about 8 hPa. Given the seasonal time scale (September–November), this can contribute significantly to regional surface climate change in spring associated with the ozone hole asymmetry change.

The SST anomaly composites demonstrate a relationship of the extreme $QSW_{min}$ longitudes with different tropical regions: western (eastern) longitudes in Fig. 7c (7f) are observed for the negative SST anomalies in the eastern (central) tropical Pacific. The easternmost longitudes in Fig. 7f demonstrate the positive SST anomalies in the western tropical Pacific and in Atlantic, similarly to the same anomaly locations in T-2m in Fig. 6b. Such positive anomaly distributions resemble the

pattern of the anomalous SST trend calculated with zonal mean trend removed (Schneider et al., 2015, their Fig. 2f) and the pattern of Atlantic Multidecadal Oscillation (Li et al., 2014; their Fig. 1e).

The SAM, QSW3 and SST patterns seen from Fig. 7 are fully reproduced with the same variables of the NCEP–NCAR reanalysis (Supplement 3, Fig. S3). This is evidence that the $QSW_{min}$

longitudes in the total ozone are steadily coupled with the climate mode (SAM) and stationary wave structure (QSW3) in the SH atmosphere, as well as with the specific SST anomaly pattern. Further evidence of this coupling is seen from Fig. 8.

Regression of the $QSW_{min}$ longitudes against the SP anomalies at grid box 1 (42.5°S, 65.0°E; Fig. 6a), a mid-latitude element of QSW3, is shown in Fig. 8a. A close similarity can be seen in

variations of the SP anomalies in grid box 1 and the $QSW_{min}$ longitudes averaged over 55–70°S: the square of the linear correlation coefficient is $r^2 = 0.57$. In Fig. 8b, time series of the $QSW_{min}$ longitude           and           the           standardized           SAM           index           (Marshall,           2003;

 are shown. The square of the linear correlation coefficient between the two time series $r^2 =$ 0.35. Therefore, approximately 57% and 35% of the $QSW_{min}$ longitude variance (both significant at the 95% confidence limit) can be explained by the surface pressure anomaly variance described by regional (grid box 1, QSW3 pattern) and hemispheric-scale (SAM) indices, respectively. This indicates that a significant interaction takes place between the identified anomalies in the tropospheric circulation and zonal asymmetry in total ozone (see Section 4 for discussion on the SH troposphere–stratosphere interaction).

As noted from Fig. 6c, the regression coefficient distribution for U200 shows meridional wave trains in the Indian and Atlantic sectors. To contrast the result of Fig. 6c, anomaly composites as in Fig. 7 but for 200 hPa are presented in Fig. 9. Similarly to Fig. 7, transition from the westernmost $QSW_{min}$ longitudes to easternmost ones (left and right columns, respectively) is accompanied by the anomaly sign reversal. The U200 anomaly composites in Fig. 9b and 9f show meridional wave train patterns over the three ocean basins, however, without clear eastward propagation, and it is difficult to determine the wave train sources. Relation of the $QSW_{min}$ longitude to these wave patterns means combined contribution of the QSW sources to the wave structure in the SH stratosphere.

It is important to note the eastward shift of the anomalies between the westernmost and easternmost $QSW_{min}$ longitudes. It can be seen comparing, for example, the QSW1 patterns in T200 at the middle–high SH latitudes (Fig. 9a and 9e) and the QSW3 patterns in W200 at the middle–sub-Antarctic latitudes (Fig. 9c and 9g). Note also more intense anomalies for the easternmost $QSW_{min}$ longitudes in Fig. 9 (right column), which demonstrate lower temperature over large part of the middle–high SH latitudes (T200, Fig. 9e), stronger sub-Antarctic zonal circulation in the lower stratosphere (U200, Fig. 9f) and enhancement of the QSW3 patterns in vertical velocities (W200, Fig. 9g) and eddy heat flux (V′T′200, Fig. 9h). Since the easternmost longitudes characterize mainly 1990s and 2000s (as outlined by solid rectangle in Fig. 5), the right column in Fig. 9 could indicate that the two decades of the strongest ozone loss were favorable for the closest interaction between the SAM/QSW3-related anomalies in the SH atmosphere and the QSW structure in total ozone. As in the case of Fig. 7, the NCEP–NCAR reanalysis shows very similar patterns in the same anomaly composites for 200 hPa (Supplement 3, Fig. S4) confirming the reliability of the results of Fig. 9.

In connection between the changes in the QSW structure in total ozone and atmospheric parameters, the clear eastward shift in the QSW3 patterns in the SH midlatitudes (Fig. 7b and 7e and Fig. 9c and 9g) is of particular interest. This tendency is analyzed in more detail using the correlative relationships between the $QSW_{min}$ longitude and air temperature.

### 3.3 Correlations between the $QSW_{min}$ longitude and the air temperature

The linear correlation between time series of the $QSW_{min}$ longitude at 65°S and the ERA-Interim air temperature was evaluated. As the QSW3 structure is concentrated in the mid- and sub-Antarctic latitudes (Figs. 6, 7 and 9), the correlation for air temperature averaged over the zone 40–60°S was calculated. In Fig. 10, the results in the longitude–height cross-section for the 36-year period 1979–2014 are presented. The sample sizes in this case mean that correlation $r = 0.27$ is significant at the 95% confidence limit (black and white contours in Fig. 10 for the positive and negative correlations, respectively).

A clear separation between the QSW1 pattern above the tropopause (peak values of $r \approx \pm 0.7$; climatological thermal tropopause shown by black curve) and the QSW3 pattern below the tropopause ($r_{max} \approx 0.5$) is seen.

The strong correlation in the stratosphere (between 200 hPa and 20 hPa, or 12 km and 26 km, respectively, in Fig. 10), firstly, demonstrates close coupling between the $QSW_{min}$ longitude and the stratospheric temperature in their interannual variations and, secondly, confirms that the $QSW_{min}$ longitude variability is mainly due to the QSW1 contribution. As could be seen from Fig. S5 in Supplement 3, the correlation using the NCEP–NCAR temperature is almost identical to the result of Fig. 10 with the ERA-Interim temperature.

An important feature of the QSW3 pattern in the troposphere is its altitudinal location: the three correlation maxima are located predominantly in the middle troposphere and their peak values are at about 500 hPa (Fig. 10). The correlation is significantly lower at 1000 hPa which could explain relatively weak SST anomalies in the SH midlatitudes in Fig. 7c and 7f. Therefore, the mid-tropospheric pressure level of 500 hPa was chosen for search of possible decadal changes in the correlations between the anomalies in air temperature and in the $QSW_{min}$ longitudes. In Fig. 11, correlation coefficient distributions in a sequence of the five 14-year intervals with the 5–6 year steps are shown. The sample size $N = 14$ and calculated lag-one autocorrelations give significance level of 95% for the correlation coefficient $r = 0.51$.

It is seen that the significant positive correlation peaks (black contours in Fig. 11) shift eastward between 1979–1992 and 1990–2003 by 30–60 degrees (thick lines in Fig. 11a–11c). Later, they remain in average at the easternmost longitudes in 1995–2008 and 2001–2014 (Fig. 11d and 11e, respectively). The QSW3 peak near 180°E exhibits a shift in opposite direction in the last time interval (Fig. 11e). Correlation sequence with the NCEP–NCAR T500 field displays similar tendencies (Supplement 3, Fig. S6).

A significant negative anomaly of the correlation coefficient ($r < -0.5$, white contours) in the South Pacific extending to polar latitudes into the Amundsen–Bellingshausen Sea low (ABSL) region is longitudinally steady (white dashed line). As noted above concerning similar anomaly location in

the RC distributions in Fig. 6a and 6b, this could be combined effect of QSW1 and QSW3. Here, meteorological variables in troposphere are correlated with the $QSW_{min}$ longitude in total ozone in their interannual variations, but do not show consistent decadal changes in spatial pattern. Note that annual cycle in the decadal shift of the ABSL longitude could contribute to distinction in zonal shift of the correlation anomalies at the middle and high SH latitudes in Fig. 11. The ABSL shifted westward in average by $-5°$/decade in September–November 1979–2008 (Turner et al., 2013, their Fig. 13) that could partly compensate eastward shift observed in the correlation maxima of the adjoining midlatitudinal zone (Fig. 11).

It can be summarized that common decadal tendencies in total ozone (Fig. 2a), QSW minimum location in the total ozone (Fig. 2b), QSW1 pattern in the lower stratosphere (Fig. 9a and 9e) and QSW3 pattern in the troposphere (Fig. 7b and 7e and Fig. 11) and lower stratosphere (Fig. 9c and 9g) exist. On interannual time scale, the SAM- and QSW1/QSW3-like patterns in the SH circulation associated with variability in the QSW minimum longitude in total ozone are dominating.

In general, the results of Section 3 show that the $QSW_{min}$ longitude changes in the TOC distribution are in statistically significant associations with both the zonal mean (SAM pattern) and regional (QSW1 and QSW3 patterns) anomalies in the spring SH atmosphere. These associations allow us to identify the SH regions, where the climate changes in spring are accompanied by ozone asymmetry changes. Our results, thus, have two differences from known impact of the spring ozone loss on the summer SH climate (Thompson et al., 2011). First, quantitative relationships (Figs. 6–11) are not built on the ozone level changes, but on a spatial parameter, which describes the changes in the location of the zonally asymmetric ozone anomaly (the $QSW_{min}$ longitude). Second, revealed climate signals characterize the Antarctic spring (September–November) and are associated with simultaneous evolution of the ozone hole asymmetry. Possible features of the troposphere–stratosphere interaction, which could contribute to occurrence of revealed links, are discussed below.

## 4 Discussion

The evolution of zonal asymmetry in Antarctic total ozone during 1979–2014 with respect to the changes in the meteorological variables in the SH troposphere and lower stratosphere has been presented in the previous section. Regressive, correlative and anomaly composite analyses show that longitudinal shift of the quasi-stationary zonal TOC minimum has a close relationship with changes in the TOC level itself and with the SAM, QSW and SST patterns. Here we discuss our results in the context of the published literature, including analysis of chemistry-climate model attribution simulations.

## 4.1 Relation between the TOC asymmetry and TOC level

Analysis covers the austral spring months September–November and the changes in the seasonally averaged longitude of the $QSW_{min}$ characterize the changes in zonal asymmetry of the ozone hole and polar vortex. Zonal asymmetry in the SH stratosphere on monthly and seasonal time scales is formed by prevailing QSW1 that propagates upward from the troposphere (Section 1). In turn, zonal asymmetry in ozone is known to be a factor influencing the planetary wave propagation from the troposphere to the stratosphere (Crook et al., 2008; Albers and Nathan, 2012). Then, the $QSW_{min}$ longitude changes could display changes in both the tropospheric wave sources (their activity and distribution) and modification of the stratospheric QSW through the feedback from zonally asymmetric ozone heating.

The results of this work show that, on interannual and decadal time scales, the $QSW_{min}$ longitudinal location is in close connection with both the $QSW_{min}$ level (Fig. 2b and 2a, respectively) and the overall ozone loss in spring (Fig. 4). Note that ozone mass deficit in Fig. 4 (black curve) characterizes the ozone hole, which is typically asymmetric relative to the South pole. Therefore, the extent of ozone depletion, which is initially depended on the wave activity, appears to be largely coupled with the ozone hole asymmetry. For example, in the years of weak (strong) wave activity in spring, large (low) ozone loss is accompanied by eastward (westward) shift of the $QSW_{min}$. This is demonstrated by the large simultaneous deviations of the $QSW_{min}$ longitude and ozone mass deficit in 1988 and 2002 (Fig. 4), the years, when enhanced wave activity resulted in the weakened polar vortices and anomalously small ozone holes (Kodera and Yamazaki, 1989; Allen et al., 2003; Varotsos, 2003b; Grassi et al., 2008).

Generally, planetary wave activity contributes significantly to the interannual variability of the ozone hole size, depth and duration (Kodera and Yamazaki, 1989; Allen et al., 2003; Varotsos, 2002, 2004; Ialongo et al., 2012). However, wave activity did not undergo so significant decadal decrease to cause the decadal tendency in ozone depletion and, therefore, can not be contributing factor to the systematic $QSW_{min}$ eastward shift. The long-term loss of Antarctic ozone is attributed to the abundance of ozone-depleting substances (ODS) peaked near the turn of the century (Salby et al., 2011; Dameris and Godin-Beekmann, 2014). Hence, evolution of ODS is initial cause of decadal change in the $QSW_{min}$ level (Fig. 2a). Related change in asymmetric ozone heating that can potentially affect the planetary wave propagation through the feedback processes (Crook et al., 2008; Albers and Nathan, 2012) could also result in the QSW structure modification. Consistent decadal tendencies seem to support assumption that evolution of zonally asymmetric ozone depletion (Fig. 2a) could be main cause of the $QSW_{min}$ shift (Fig. 2b). Note that decadal changes of the ozone hole metrics seem to be also influenced by ozone depletion itself. For example, the delay

in the final warming of the Antarctic vortex and seasonal disappearance of the ozone hole in 1980s–1990s appears influenced by increasing overall ozone loss (Haigh and Roscoe, 2009).

Thus, interaction between the planetary waves and ozone could underlay observed evolution of the asymmetric ozone hole: (i) zonal asymmetry in Antarctic ozone is initially induced by the QSW propagating from the troposphere, (ii) significant change in the TOC level occurs within asymmetric ozone hole causing change in zonally asymmetric ozone heating that, (iii) through the feedback processes, could result in the QSW structure modification.

## 4.2 The SAM pattern

As it is known, positive SAM polarity is associated with the enhanced westerlies around Antarctica and decreased surface pressure and air temperature in the polar region (Thompson and Wallace, 2000). The eastward shift of the QSW minimum in total ozone is accompanied by similar indications of the positive SAM polarity: strengthening of zonal wind around Antarctica (U-10m in Fig. 7d and U200 in Fig. 9f), surface pressure decrease (SP in Fig. 7e) and air temperature decrease (T-200 in Fig. 9e and T500 in Fig. 11) in the SH high latitudes.

Our results do not give information on the direction of the 'QSW$_{min}$–SAM' coupling. Tropospheric circulation disturbances can influence the evolution of the stratospheric polar vortex and, in turn, stratospheric processes can induce a tropospheric response that projects on the annular mode (Thompson ans Wallace, 2000). It is known that the Antarctic ozone losses in spring impact the SAM-related tropospheric circulation predominantly in summer (Thompson et al., 2011; Schneider et al., 2015) and the spring SAM index shows near-zero decadal trend (Marshall, 2003; Fogt et al., 2009; Arblaster and Gillett, 2014). In addition, in our relationships for zonal wind (Fig. 7a and 7d and Fig. 9b and 9f), it is difficult to reveal evidence of the poleward shift of the westerly jets observed usually in the positive SAM polarity (Thompson and Wallace, 2000). This is because of strong disturbance of zonal anomaly orientation by the midlatitude QSW3 structure. So, on decadal time scale, influence of the ozone change accompanied by the QSW$_{min}$ longitude change (Fig. 2a and 2b, respectively) on the spring SAM pattern can not be identified from our results.

Nevertheless, such feedback possibility can not be fully excluded, at least on interannual time scales. As shown by Son et al. (2013), stratospheric ozone concentration in September is strongly correlated with the SAM index in October with $r = -0.7$. The mechanism of this time-lagged downward coupling remains to be determined.

Thus, the appearance of the SAM-like patterns in our relationships (Fig. 6, Fig. 7 and Fig. 9) can be interpreted in two ways that are influenced by interannual variability: first, the strength and pattern of the circulation in the spring SH troposphere can conceivably play a significant role in determining the location of the QSW structure in the stratosphere, and, second, the QSW structure

in the stratosphere can potentially provide a downward influence on the tropospheric circulation. This could mean that assumed feedback processes of the asymmetric ozone loss (noted in Section 4.1 with respect to the QSW structure in the stratosphere) may spread into the troposphere in the springtime. The results by Son et al. (2013) demonstrate such possibility using ozone index for the SH polar area. However, ozone variability in the SH polar area is typically combined with the ozone asymmetry variability (Fig. 2) and combined effect of the ozone itself and ozone asymmetry could affect the interannual variability of SH surface climate in spring noted in (Son et al., 2013).

**4.3 The QSW1/QSW3 patterns**

Unlike the SAM pattern, the atmospheric QSW3 pattern in the midlatitude troposphere demonstrates long-term changes (Fig. 11) consistent with both the ozone loss tendency (Fig. 2a and Fig. 4, black curve) and the $QSW_{min}$ longitude shift (Fig. 2b). Positive correlations of up to $r = 0.7$–$0.8$ and the longitudinal shift of the three correlation maxima in Fig. 11 are evidence of significant coupling between $QSW_{min}$ and QSW3 on both interannual and decadal time scales. QSW3, although smaller than QSW1, is a dominant feature of the SH midlatitude circulation on daily, seasonal and interannual timescales (Raphael, 2004, and references therein). The QSW3 ridges determined from the 500-hPa geopotential height anomalies over the period 1958–2001 by Raphael (2004) are located climatologically over southern South America, the southern Indian Ocean and southwest of New Zealand.

Those ridge locations correspond to the correlation maxima in Fig. 11a and 11b for the periods 1979–1992 and 1984–1997, respectively, which cover the last two decades of the time interval in Raphael (2004). The largest eastward shift of the QSW3 pattern occurred between the 1980s and 1990s (Fig. 11a and 11c, respectively). The central ridge that is located on average near 180°E (Fig. 10) drifted from the southwest of New Zealand in the 1980s (as in Raphael (2004) and shown in Fig. 11a) to southeast of New Zealand by the early 2000s (Fig 11d), covering over about 60° of longitude. Apparent in the longitude sector of this ridge in the 1990s and 2000s has been a cooling of the tropical central Pacific that has been linked to wind-driven vertical circulation changes in the Pacific Ocean (England et al., 2014); this process is also associated with the spin-up of the subtropical gyres and associated surface ocean warming (see Fig. 2 of England et al. (2014)) in the region of the ridge. In the 2000s and 2010s, as seen from Fig. 11c–11e, the ridge locations showed less drift. In particular, Figure 11e indicates that the central ridge has drifted west back towards New Zealand.

Note that in the years of maximum ozone hole area (easternmost $QSW_{min}$), the midlatitude wave 3 anomalies of the positive correlation partly cover New Zealand and southern tip of South America (Fig. 11c–11e, and Fig. S6c–S6e). Positive anomaly here corresponds to climate warming in the

years of the easternmost QSW$_{min}$ migrations. In future, predicted ozone recovery may be accompanied by further westward shift of the wave 3 pattern and by weakening of the positive anomaly influence in region of New Zealand and South America (similarly to Fig. 11a and Fig. S6a).

As seen from Fig. 11a and Fig. S6a, both reanalyses show negative correlation anomalies over Australia and East Antarctica in the first time interval 1979–1992 (pre-ozone hole and first ozone hole years, westernmost QSW$_{min}$,). Later, these negative anomalies weaken (Fig. 11b–11d) and appear again in the latest time interval 2011–2014 (Fig. 11e). Note that regression in Fig. 6b and correlation in Fig. S2b also show negative anomaly over south-east part of Australia. These tendencies indicate that Antarctic ozone recovery to pre-ozone hole level may be accompanied by strengthening of negative coupling 'tropospheric temperature – QSW$_{min}$ longitude' in this region. All of these climate effects need further analysis.'

The QSW1 structure in the lower stratosphere covers the middle and high SH latitudes, and, in the two reanalyses, shows consistent eastward shift between the westernmost and easternmost locations of the QSW$_{min}$ (Fig. 9a and 9e; Fig. S4a and S4c). Note that the QSW1 pattern in Fig. 10 takes the altitude range where vertical ozone profile in the austral spring undergoes the largest ozone decrease (Chubachi, 1984; Varotsos, 2003b; Solomon et al., 2005). Dynamical activity changes the extent of ozone depletion from year to year and, in the conditions of zonal asymmetry of the polar vortex, enhances the TOC variability. The role of zonal asymmetry becomes more important due to vertical non-alignment of the vortex structure (Varotsos, 2004). The vortex appears progressively shifted with height towards South America and high ozone in the middle stratosphere (around 30 km) within the Australian sector masks ozone depletion in the lower stratosphere (Kondragunta et al., 2005; Tully et al., 2008) resulting in additional variability of stratospheric temperature, column ozone and ozone hole characteristics. In this way, vertical changes in the QSW1 structure impact the ozone profile over Antarctica and contribute to the strong correlation between zonal asymmetry in ozone distribution (QSW$_{min}$ longitude) and stratospheric temperature in Fig. 10.

Overall, the long-term shifts in the QSW3 centers and the QSW1 pattern show a temporal evolution that is qualitatively similar to decadal changes in both ozone depletion (Fig. 2a and Fig. 4, black curve) and the shift of QSW$_{min}$ in total ozone (Fig. 2b). This similarity suggests that decadal-scale changes in the QSW structures in the troposphere and stratosphere could either have independent common source, or result from troposphere–stratosphere interaction in the SH extratropics including feedback processes assumed in Sections 4.1 and 4.2. Both possible mechanisms deserve further analysis using observational data and models.

Simulated influence of the greenhouse gases increase on the eastward phase shift in the stratospheric stationary waves (Wang et al., 2013) indicates the possible contribution of the first

mechanism. This model simulation links eastward phase shift to the strengthening of the subtropical jet driven by greenhouse gases forcing via sea surface warming. Induced eastward shift is projected by the model to the end of the 21$^{st}$ century (Wang et al., 2013).

Agosta and Canziani (2011) have shown that there are significant interactions/coupling between the ozone layer, the troposphere, and the stratosphere during the austral spring, which can be traced by the phase changes in TOC and QSW1 in the stratosphere. Such changes and troposphere–stratosphere interactions, by Agosta and Canziani (2011), are linked to both the upward and downward propagation of quasi-stationary wave anomalies. Taking into account the results by Son

et al. (2013) and the results of our work, the second of the mentioned mechanisms that suggests contribution of ozone change may be more effective in the recent decades.

## 4.4 The SST patterns

Clear difference between the SST anomaly patterns for the westernmost and easternmost QSW$_{min}$

locations has been identified using the two reanalysis data (Fig. 7c and 7f; Fig. S3c and S3f). Particularly, the westernmost locations show the negative SST anomaly in the eastern tropical Pacific (La Niña type anomaly; more intense in the NCEP–NCAR reanalysis than in ERA-Interim, Fig. S3c and Fig. 7c, respectively). The westernmost locations show the negative SST anomaly in the central tropical Pacific (Fig. 7f and Fig. S3f).

Difference between the two regions of the tropical Pacific in their coupling with the QSW structure in the SH stratosphere was noted by Lin et al. (2012). By Lin et al. (2012), westward QSW phase shift is seen for negative SST anomalies (La Niña events) in the eastern Pacific and eastward shift is seen for warm SST anomalies in the central Pacific. The results of Fig. 7c and Fig. S3c show a similar association between the westernmost QSW$_{min}$ longitudes and negative SST anomaly in the

eastern Pacific. However, the easternmost longitudes in Fig. 7f and Fig. S3f show also a negative anomaly in the central Pacific, as distinct from a positive one (Lin et al., 2012).

In the individual years, large westward phase shift in the QSW$_{min}$ occurred in 1988 and 2002 (Fig. 2b and Fig. 4). Sources of anomalous planetary waves in these years have been identified in both the tropical Pacific (Kodera and Yamazaki, 1989; Grassi et al., 2008) and the SH extratropics

(Nishii and Nakamura, 2004; Peters et al., 2007). It has been noted that the evolutions of sea surface temperatures in the tropical Pacific in the spring months of 1988 and 2002 were different (Varotsos et al., 2003b) with strong La Niña and emerging El Niño conditions, respectively (McPhaden, 2004; see also time series for the indices Niño 3 and Niño 4 at http://www.esrl.noaa.gov/psd/data/climateindices/).

In the case of 2002 this is counter to the expectation from Lin et al. (2012) based on the prevailing positive central Pacific SST anomaly (McPhaden, 2004) and disagrees with Fig. 7f and Fig. S3f, since a negative SST anomaly exists in this region.

Note that Fig. 7f and Fig. S3f show a significant negative SST anomaly in the South Pacific and positive SST anomalies in the western tropical Pacific and in the Atlantic. Similarly to the PSA

mode in the Pacific sector, poleward propagating Rossby wave train could be driven by the Atlantic SST anomalies (e.g., Li et al., 2014). Combined influences of the wave train could result in other phase shift direction in the SH stratosphere planetary waves than from positive anomaly in the central tropical Pacific in (Lin et al., 2012). In particular, the SH extratropical Rossby wave activity that propagated into the stratosphere in the spring of 2002 (Nishii and Nakamura, 2004; Peters et

al., 2007) could also have contributed to the observed $QSW_{min}$ shift in that year.

In general, the results of Section 3 reveal indications of the connection between changes in zonal asymmetry in Antarctic total ozone and changes in zonally symmetric (SAM) and zonally asymmetric (QSW1 and QSW3) patterns in the SH circulation, as well in the SST patterns. These results are, in general, in agreement with known evidence of coupling between Antarctic ozone and

SAM (Thompson and Wallace, 2000; Waugh et al., 2009; Thompson et al., 2011), the SH QSW structure (Agosta and Canziani, 2011; Wang et al., 2013) and the SSTs (Kodera and Yamazaki, 1989; Grassi et al., 2008). These couplings allow us to identify the SH regions, where the climate changes in spring are accompanied by the ozone asymmetry changes.

**4.5 Attribution of longitude shift**

We now turn to chemistry-climate model simulations to further examine long-term changes in the QSW pattern. Here we consider simulations for the Chemistry-Climate Model Initiative (CCMI) (Eyring et al., 2013) produced by the chemistry-climate version of the Australian Community Climate and Earth System Simulator (ACCESS-CCM; Stone, 2015; Stone et al., 2016).

Specifically, we examine single simulations from the REF-C2, SEN-C2-fODS and SEN-C2-fGHG scenarios which are described in detail by Stone (2015). The REF-C2 simulation covers the period 1960–2100 and includes evolving concentrations of greenhouse gases (GHGs) and Ozone Depleting Substances (ODSs); the SEN-C2-fODS and SEN-C2-fGHG simulations are similar to REF-C2 except that ODSs and GHGs, respectively, are separately fixed at 1960 levels.

The ACCESS-CCM simulations favorably reproduce general characteristics of ozone depletion and stratospheric stationary waves compared with observations and various similar models (Stone, 2015; Stone et al., 2016). In Fig. S7a of Supplement we show the meridional cross-section of the correlation between temperature and $QSW_{min}$ longitude for the REF-C2 simulation over 1979–2014 for direct comparison with Fig. 10. Although the strength of the correlations in the REF-C2

simulations (Fig. S7a) is weaker than seen in the ERA-Interim reanalysis (Fig. 10) and in the NCEP–NCAR reanalysis (Fig. S5), the simulation shows similarly located patterns; for example, the region of positive correlation in the lower stratosphere is of similar extent as in the reanalyses, but shifted ~30° westward for the REF-C2 simulation. Correlation analysis for an additional model simulation, REF-C1, over 1979–2010, which has a similar setup to REF-C2 (Stone, 2015), shows in Fig. S7b a closer agreement with Fig. 10 and Fig. S5 and we conclude that model internal variability has a noticeable influence on the correlations on these timescales.

Stone (2015) has investigated the long-term shift in the Southern Hemisphere TOC patterns in spring and summer in relation to GHG and ODS changes, where the phase of the wave-1 component of the TOC (obtained from a zonal Fourier decomposition) was analyzed in a similar manner to Grytsai et al. (2007a). By regressing the wave phase against measures of GHG and ODS forcing, Stone (2015) finds that ODS forcing explains a significant fraction of the long-term variability in the wave-1 longitude compared with GHG in spring at 50°S and 60°S (his Fig. 5.6), with the strongest influence at the equatorward latitude. From examining also the wave-1 pattern in temperature at 50 hPa and 500 hPa and zonal wind at 10 hPa, Stone (2015) concludes that temperature changes associated with the spring ozone depletion play the most important role in the eastward shift of the TOC pattern at these latitudes, whereas influences from GHG changes are mitigated by a blocking effect from the continental features, particularly the Andes. As noted in Section 4.3, simulated GHG increase during the 21$^{st}$ century shows the eastward phase shift in the stratospheric stationary waves and this shift is linked to the strengthening of the subtropical jet driven by greenhouse gases forcing via sea surface warming (Wang et al., 2013).

While Stone (2015) only examined the wave-1 phase, we show in Fig. 12 long-term trends in QSW$_{min}$ from the REF-C2, SEN-C2-fODS and SEN-C2-fGHG simulations averaged over 55°S to 70°S, as well as the observational data presented in Fig. 5. While there is large interannual variability, the SEN-C2-fGHG run (which has evolving ozone changes and fixed GHG level), and to a lesser extent the REF-C2 run (which has both evolving GHG and ODS changes) both show a tendency for an eastward shift (peaking around 2000–2025) and then westward relaxation over the ozone hole period (to about 2060). On the other hand, the SEN-C2-fODS simulation in which GHGs evolve, shows a much smaller trend and overall less variability over the ozone hole period compared with the other simulations. We note that while the observations show a mean phase shifted ~30–40° further east than the simulations (similar to the difference noted between the positive correlation in the lower stratosphere between Figs. 10 and S7), the variability in the observational record is not dissimilar to the variability on similar time scales in the simulations.

Overall we conclude from the simulations that ozone depletion is providing a strong influence on the QSW$_{min}$ phase in the current period, and may be playing an important role in the level of

interannual variability in the stratospheric quasi-stationary wave pattern, and possibly also the
       pattern in the troposphere, at least over latitudes around 60°S.

## 5 Conclusions

       We have examined the variability of the minimum in the quasi-stationary pattern in total column
ozone in spring at high southern latitudes using observations and model simulations. Our main
       results are:

       1) In interannual variations, the longitude of the QSW minimum in total ozone is in close
       association with the SAM index, the QSW1/QSW3 patterns in the meteorological variables and the
       SST patterns. Particularly, easternmost (westernmost) longitudes of the QSW minimum are
accompanied by shift of the SAM index to positive (negative) polarity (Fig. 8b).

       2) On the decadal time scale, a consistency between the longitudinal shifts of the QSW minimum in
       total ozone (Fig. 2b) and the QSW3 pattern in the mid-tropospheric temperature (Fig. 11) has been
       shown.

       3) The SST anomalies over the Pacific and Atlantic basins contribute to the variability in the QSW
minimum location.

       4) Based on the results from our attribution experiments with the ACCESS-CCM climate model,
       increased levels of ODS and GHGs both tend to shift the $QSW_{min}$ eastward. From the simulations,
       asymmetric ozone depletion in spring is likely having a strong influence on the phase of the
       stratospheric wave pattern around 60°S.

Regression, correlation, anomaly composite and model analyses show that longitudinal variability
       in location of the quasi-stationary zonal TOC minimum has a close relationship with variability in
       the TOC level itself, in the SAM/QSW patterns in the meteorological variables and in the SST
       patterns. Therefore, these couplings allow us to identify the SH regions, where the climate
       variability and climate change in austral spring could be accompanied by the ozone hole asymmetry
changes. Related shifts in the tropospheric QSW3 pattern play a role in climate variability in
       regions of Australia, New Zealand and southern tip of South America. On the one hand, the results
       suggest combined influences of the QSW sources on the stationary wave structure in the SH
       stratosphere and, on the other hand, they indicate possible ozone change feedback affecting the
       wave structure.

Chemistry-climate models predict that the Antarctic ozone will return to the 1980 level in the
       second half of the 21[st] century, in 2050–2070 period (Siddaway et al., 2013; Dameris and Godin-
       Beekmann, 2014; Solomon et al., 2016) and the period of ozone recovery will take approximately 3
       times longer than did the growth epoch of the ozone hole (approximately 2000–2060 and 1980s–
       1990s, respectively). From our simulations, recovery in ozone levels during the next 2–3 decades

will allow the $QSW_{min}$ location to generally reverse direction and drift westwards, until the forcing by increasing GHGs starts to dominate and eastward drift resumes (after approximately 2060). The combined impacts of the decadal changes in ozone and GHG on the SH stationary waves indicated by our results suggest that surface influences can be expected over the remainder of this century. It is also possible that interannual variability in the stratospheric stationary waves may reduce as

ozone depletion subsides, and this may also have a bearing of interannual variability in high latitude surface climate. Further model evaluations are needed to assess these indications.

## Author contribution

Based on ideas developed by G. P. M., A. V. G. performed data development and provided analysis

with contribution from G. P. M., A. R. K., O. M. E and K. A. S. G. P. M., A. R. K. and O. M. E. provided additional explanation of the outputs. A. V. G. prepared the manuscript with contributions from G. P. M., A. R. K. and O. M. E.

**Acknowledgments.** Authors thank the two anonymous Referees for their comments and helpful

suggestions. TOMS and OMI daily total ozone data provided by the Ozone Processing Team, NASA Goddard Space Flight Center, USA, from their Web site at http://ozoneaq.gsfc.nasa.gov. Multi-sensor reanalysis data from http://www.temis.nl are used. Data on ozone mass deficit in the Southern high latitudes are from http://ozonewatch.gsfc.nasa.gov. NCEP Reanalysis data provided by the NOAA/OAR/ESRL PSD, Boulder, Colorado, USA, from their Web site at

http://www.esrl.noaa.gov/psd/. The ERA-Interim data from the European Centre for Medium-Range Weather Forecasts reanalysis at http://www.ecmwf.int/en/research/climate-reanalysis/era-interim were used. This work was partly supported by Taras Shevchenko National University of Kyiv, project 16BF051-02, and by the Polar FORCeS project no. 4012 of the Australian Antarctic Science Program, the Australian Research Council's Centre of Excellence for Climate System

Science (CE110001028), the Commonwealth Department of the Environment (grant 2011/16853), and National Institute of Water and Atmospheric Research as part of its New Zealand government funded, core research, and by the Marsden Fund Council from government funding, administered by the Royal Society of New Zealand (grant 12-NIW-006). This research was undertaken with the assistance of resources provided at the National Computational Infrastructure National Facility

systems at the Australian National University through the National Computational Merit Allocation Scheme supported by the Australian Government.

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

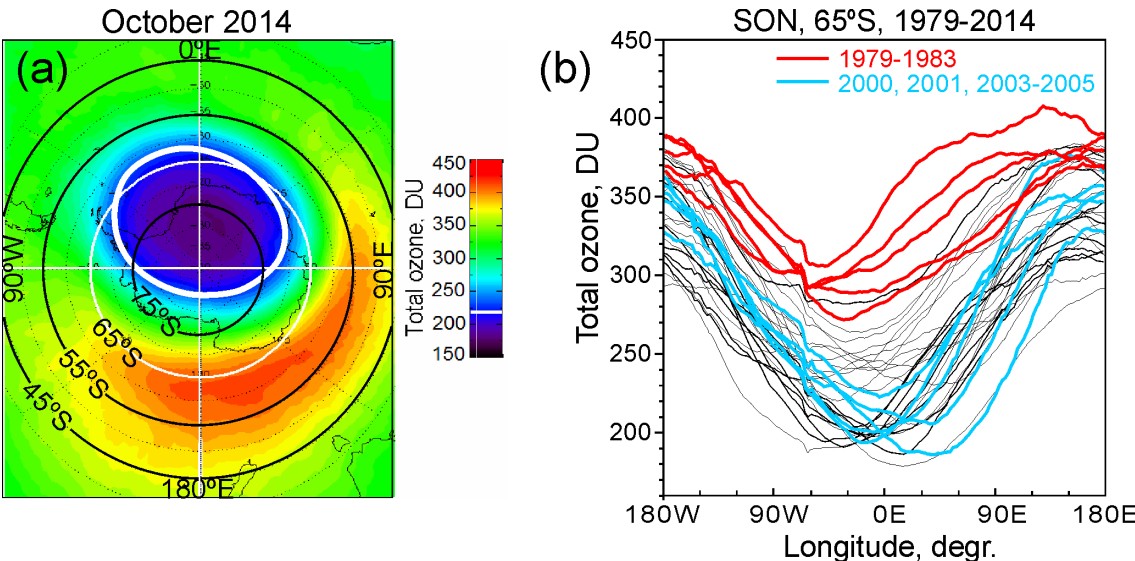


**Figure 1:** (a) Total ozone in the SH middle/high latitudes in October 2014. (b) Zonal dependence of the total ozone at 65°S averaged for September–November for each year between 1979 and 2014; Red curves show pre-ozone hole years 1979–1983 and blue curves show the years of maximum ozone hole 2000–2005 (apart from the anomalous 2002). Modified from Grytsai et al. (2007a).


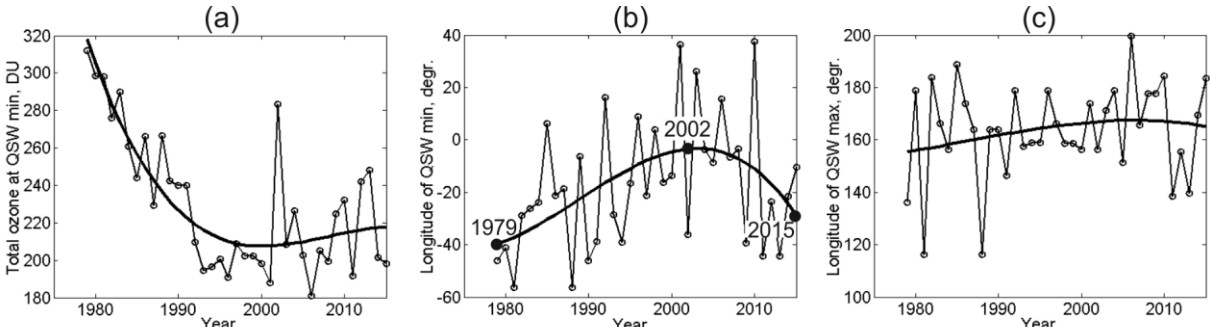

**Figure 2:** (a) Total ozone at zonal QSW minimum, (b) longitude of QSW minimum and (c) 980 longitude of the QSW maximum at 65°S averaged for September–November. Thin lines are time series of 1979–2015 and thick lines are polynomial fits of degree 3. Longitudes for 1979, 2002 and 2015 from polynomial fitting are marked in (b) by closed circles.


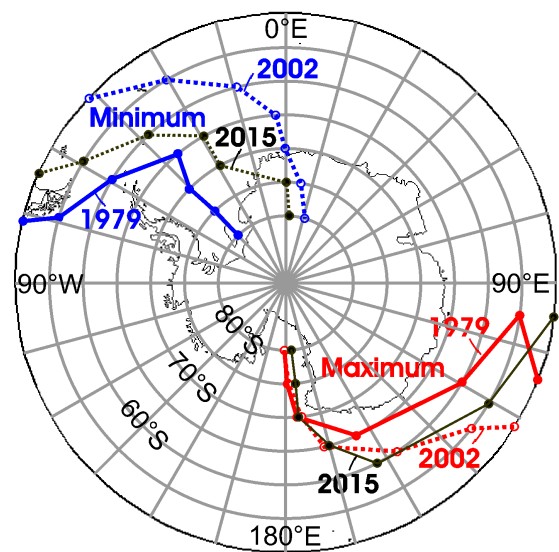

**Figure 3:** Map of longitudinal locations of zonal QSW maximum (red) and zonal QSW minimum (blue) at seven latitudes between 50°S and 80°S; westernmost (easternmost) longitudes in 1979 (2002) determined from the polynomial fit, as shown in Fig. 2b, relate to thick (thin) lines. Black dotted line marks longitudes for 2015. Modified from Grytsai et al. (2007a).

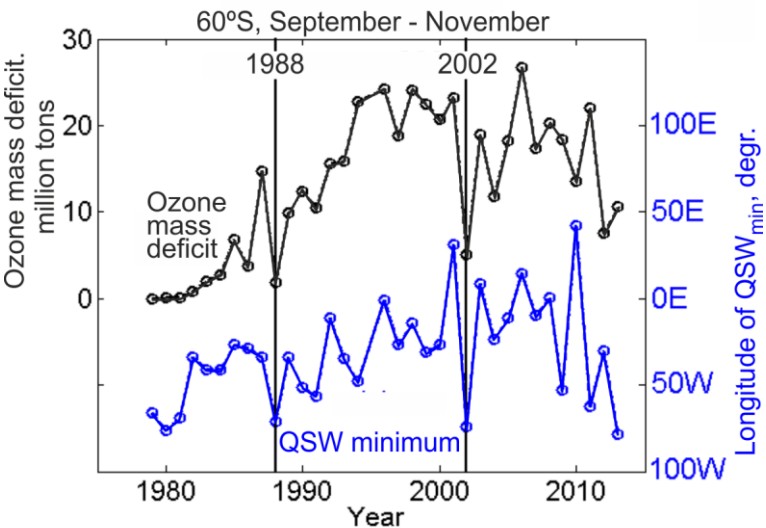

**Figure 4:** Ozone mass deficit and position of the quasi-stationary minimum in the ozone distribution at 60°S averaged over September–November. Pearson correlation coefficient between the two time series is $r = 0.57$. Data for the ozone mass deficit in the SH high latitudes are from http://ozonewatch.gsfc.nasa.gov/meteorology/SH.html. Years of large (1988) and extreme (2002) stratospheric warmings are indicated by vertical lines.

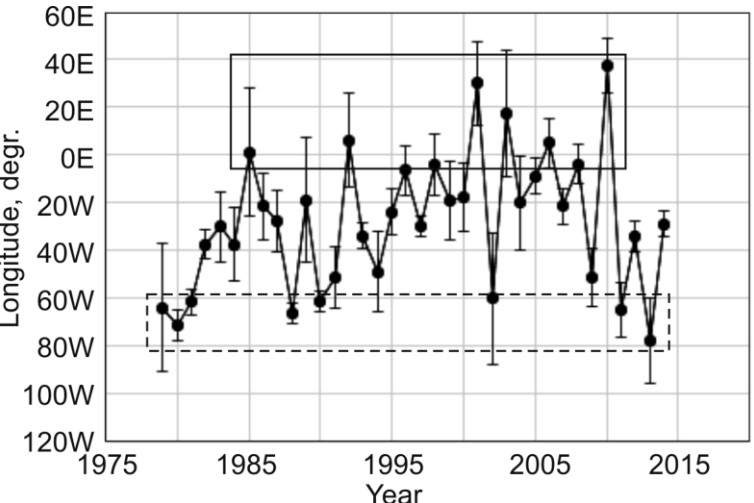

**Figure 5:** Longitude of MSR TOC ozone minimum (QSW$_{min}$) for SON averaged over four latitudes between 55°S and 70°S, inclusive. The vertical bars span ±1 standard deviation of the individual values used in each average. The solid (dashed) rectangle outlines the upper 80[th] percentile (lower 20[th] percentile) corresponding to the maximal eastward (westward) shift of the QSW$_{min}$ longitude.

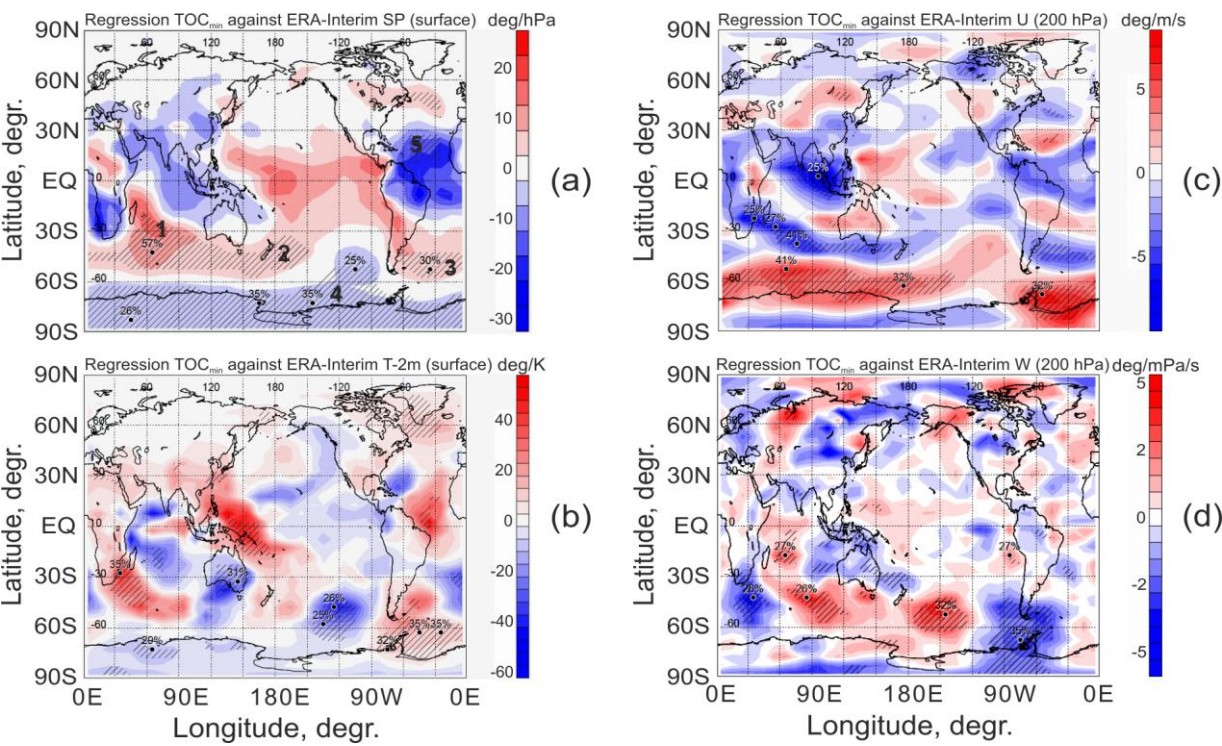

**Figure 6:** Regression coefficient of the TOC QSW minimum longitude against ERA-Interim climatological anomalies of (a) surface pressure (SP), (b) 2 metre air temperature (T-2m), (c) 200-hPa zonal wind speed (U200) and (d) 200-hPa vertical pressure wind speed (W200) for SON 1979–2014. The units are degrees of longitude per: (a) hPa, (b) K, (c) m s$^{-1}$ and (d) mPa s$^{-1}$. The hatching is shown where the correlation between the two variables in each panel is significant at the 95% confidence limit.

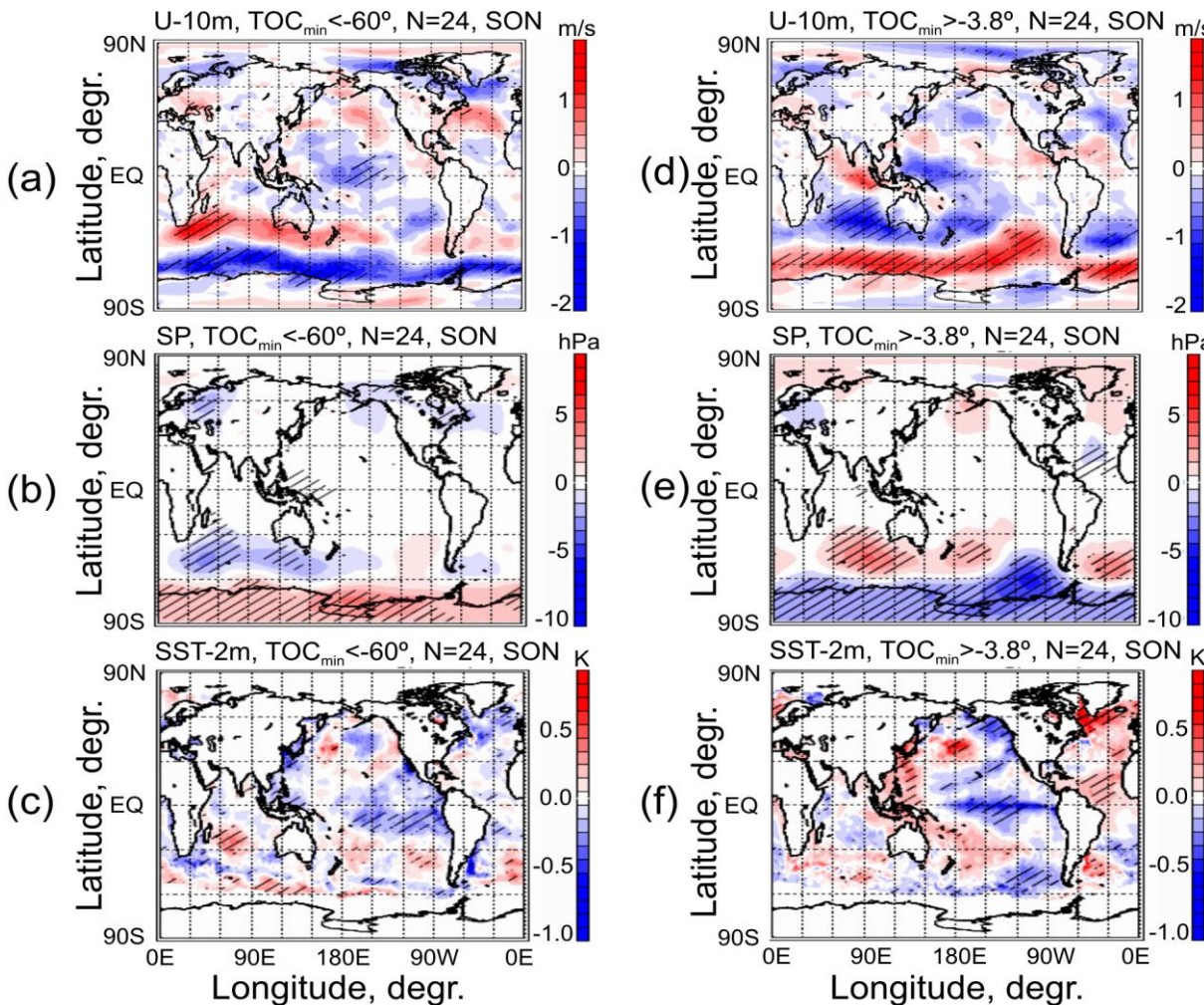

1020

**Figure 7:** Anomaly composites with respect to the mean climatology for 1979-2014 of ERA-Interim surface meteorological variables for (left) the lower 20[th] percentile of mean SON $QSW_{min}$ longitudes (western phases) and (right) the upper 80[th] percentile of mean SON $QSW_{min}$ longitudes (eastern phases). Rows (from top to bottom) are (a, d) 10-m zonal wind (U-10m in m s$^{-1}$), (b, e) surface pressure (SP in hPa) and (c, f) sea surface temperature (SST in K). Diagonal shading indicates regions significant at the 95% confidence limit (evaluated by comparing the value of each grid box with its standard deviation).

1030

1035

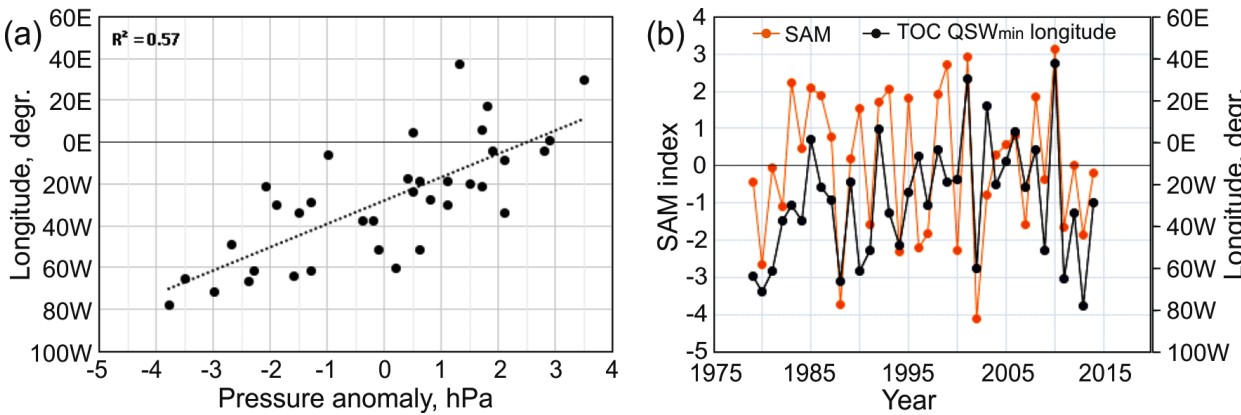

**Figure 8:** (a) Regression of surface pressure anomaly for grid box 1 (42.5°S, 65.0°E) of Fig. 6a (horizontal axis) against the $QSW_{min}$ longitude (vertical axis). (b) Time series of the standardized SAM index (orange, left axis) and the $QSW_{min}$ longitude (black, right axis). The time series of the $QSW_{min}$ longitude is that shown in Fig. 5. All variables are averaged over September–November. The $R^2$ value is significant at the 95% confidence limit.

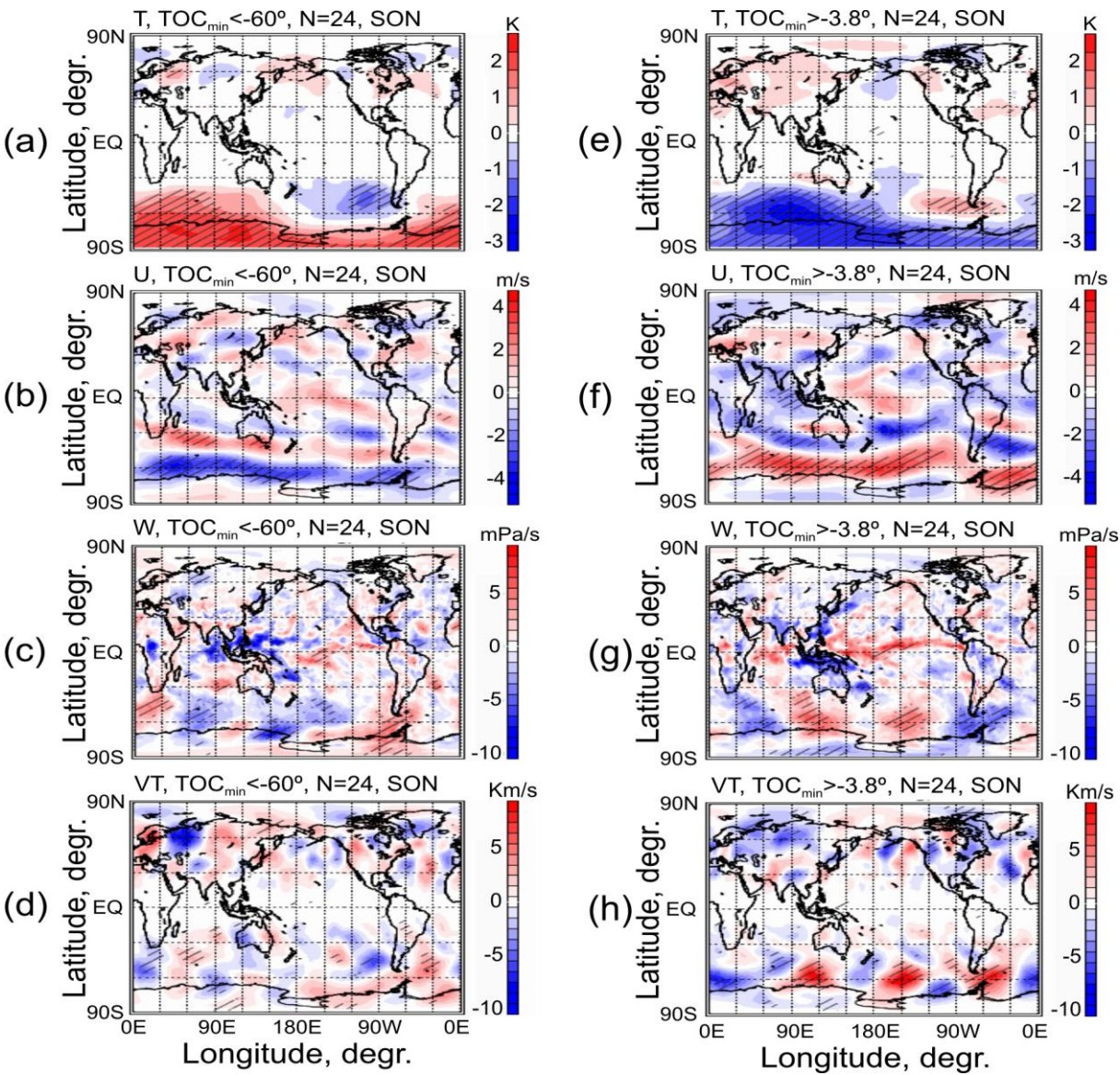

**Figure 9:** As in Fig. 7, but for ERA-Interim meteorological variables at 200 hPa: (a, e) air temperature (T200 in K), (b, f) zonal wind (U200 in m s$^{-1}$), (c, g) vertical pressure wind (W200 in mPa s$^{-1}$), and (d, h) eddy heat flux (V′T′ in K m s$^{-1}$). Diagonal shading indicates as in Figure 7.

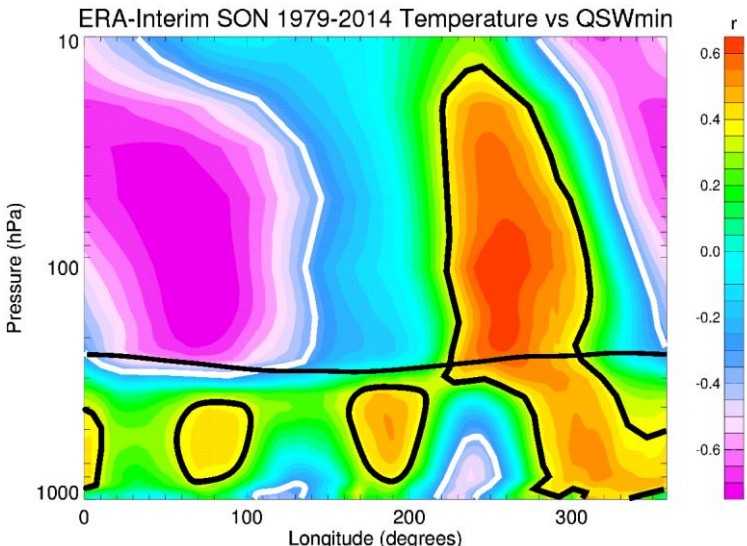

**Figure 10:** Longitude–height cross-section of the correlation between the QSW$_{min}$ longitude at 65°S and ERA-Interim air temperature averaged over the zone 40–60°S for SON 1979–2014. Thick black curve marks climatological thermal tropopause. Black (white) contours show positive (negative) correlations significant at the 95% confidence limit.

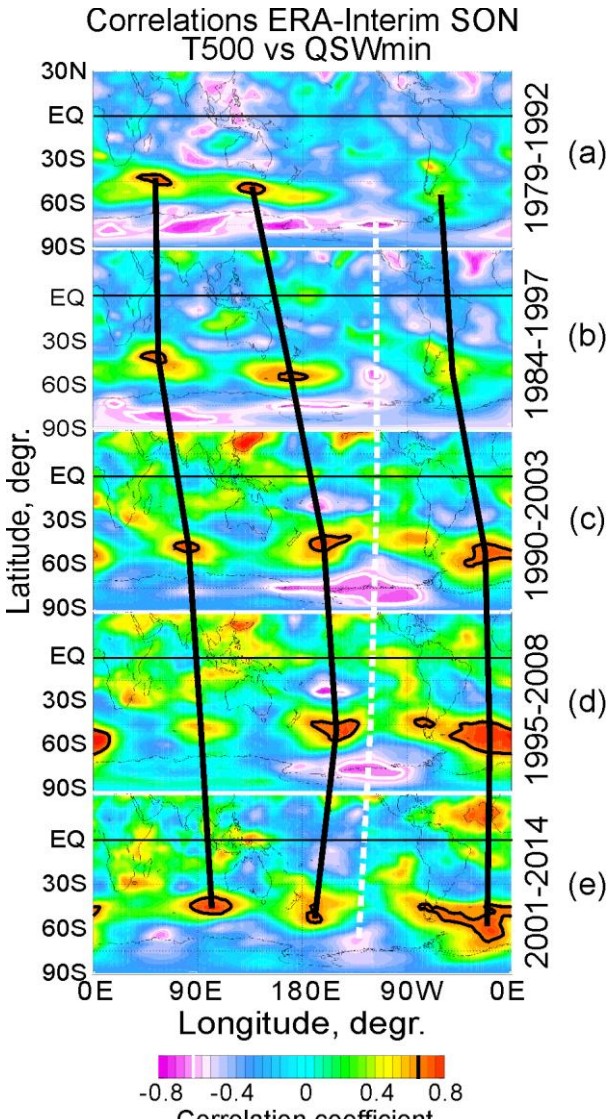

**Figure 11:** Correlation between the $QSW_{min}$ longitude at 65°S and ERA-Interim air temperature at 500 hPa south of 30°N. Five sequential 14-year intervals with 5-6-year step are presented. Black (white) contours show positive (negative) correlation peaks $r = 0.65$ significant at the 99% confidence limit. Thick solid black (dashed white) lines mark mean longitudinal positions of the positive (negative) correlation peaks in the QSW3 structure.

1080

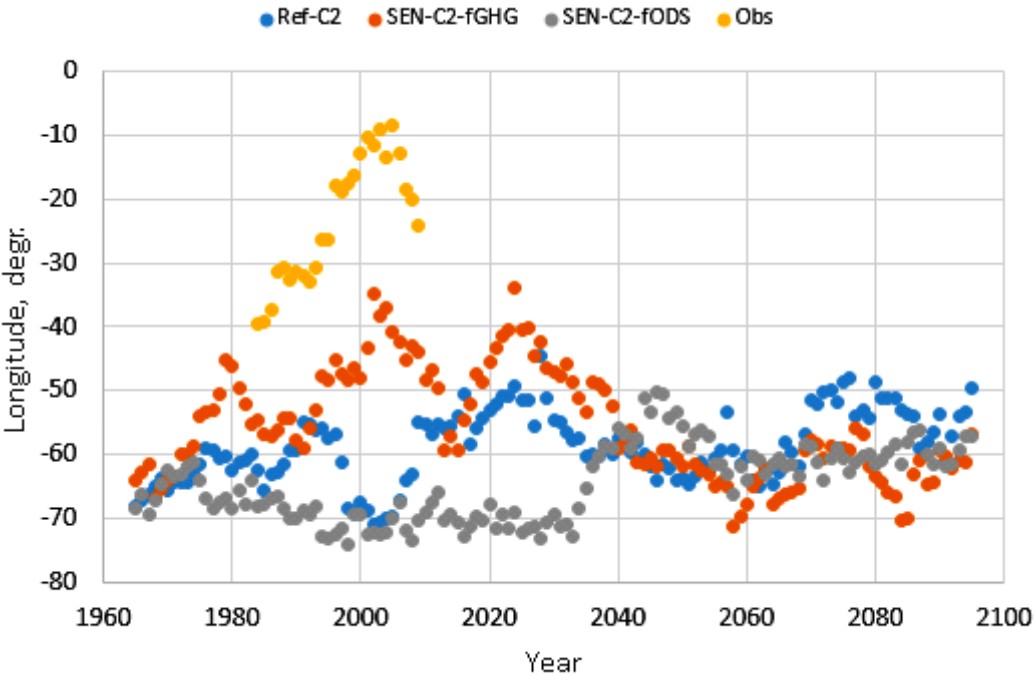

**Figure 12:** Variation of QSW$_{min}$ longitude from the ACCESS-CCM model runs (blue: REF-C2, orange: SEN-C2-fGHG, grey: SEN-C2-fODS) compared with the observations (yellow) shown in Fig. 5. For ACCESS-CCM, the QSW$_{min}$ longitude was obtained by averaging the results for four latitude bands of 5° width between 55°S and 70°S during SON. All time series have been smoothed with an 11 year running mean (i.e. extending from –5 years to +5 years).

1085