# Peer review of "Evolution of the eastward shift in the quasi-stationary minimum of the Antarctic total ozone column"

_Atmospheric Chemistry and Physics, 2016_

## Referee Comment (RC1) · Anonymous Referee #1 · 19 Aug 2016

The paper presents new results that are useful for a better understanding of the spatio-temporal dynamics of the Antarctic ozone depletion. Therefore, the starting point for its publication is very positive.

However, missing cited literature on crucial thematic topics mentioned in the paper create negative impression to the reader. As an example, the citations given for both the ozone hole and the first major sudden stratospheric warming over Antarctica in 2002 are not adequate (see below my detailed comments).

In addition, discussion on the plausible contribution of El Nino events in 1988 and 2002 should be made. For instance, the existing literature on this subject suggests that El Nino characteristics in 1988 and 2002 may not have been similar. Brief discussion on

[Figure]

it should be added.

Furthermore, it has been suggested often that the delayed SSWs in Antarctica are being directly and primarily caused by the ozone destruction by CFCs. It would be worthwhile short discussion on this for the years 1988 and 2002 to be incorporated.

Finally, comments on the discrepancies observed in the ozone vertical distribution alongwith TOC variability would be very informative. For example, the cases in 2001 and 2002 would be very interesting to be added.

A few additional points of interest are:

Page 1, L 34-35: There are earlier papers on the field and should be cited, e.g.:

Chubachi, Shigeru. Preliminary result of ozone observations at Syowa station from February 1982 to January 1983.ÂăMemoirs of National Institute of Polar Research. Special issueÂă34 (1984): 13-19.

Page 1, L 37-38: This has already been published 20 years before Solomon et al., (2005). See for instance Chubachi, (1984) and read the fourth result in the Summary of this paper and the discussion presented in it.

Page 2, L 48-53: The fundamental citations on the field of first major sudden stratospheric warming over Antarctica in Sept 2002 and the vortex split are missing.

Page 3, L 123-124: In both formulas insert space between m$\lambda$ and d$\lambda$

Page 5, L 174: Add a note that r stands for the correlation coefficient and briefly discuss its statistical significance
* * *

---

## Referee Comment (RC2) · Anonymous Referee #2 · 9 Sep 2016

Review of the manuscript "Evolution of the eastward shift in the quasi-stationary minimum of the Antarctic total ozone column" by

Grytai, Milinevsky, Kiekociuk and Evtushevsky

In the publication, the authors describe the observation of an eastward shift of the minimum of the Antarctic total ozone column. After stating the effect, the authors proceed to search for correlations and relationships with several other changes in meteorological fields.

The authors start with a literature review and describe briefly the methods they used. It is followed by detailed description of a lot of phenomena and a Conclusion and sum-

mary.

General

I have been often left wondering, what the new work in the publication is, where existing work is just continued and, most important, how relevant the work is. Mostly the authors describe phenomena and correlations. The authors frequently jump from finding a correlation of two features to the conclusion, that the connection is shown. Examples will be named below. I also found contradictionary statements, see also below.

Although the authors used models (ERA INTERIM and NCEP) they did not investigate, if the phenomena they describe are reproduced by the models. At least ERA INTERIM provides ozone also. Although they are assimilation models, it would be crucial to investigate if they reproduce the correlation described here also. The authors also failed to explain, why two models have been used. I would expect this part of the Section 2.

I understand that this is not a modelling paper and am not asking the authors to evaluate models. However, easily accessible data should have been used to help the reader to classify the paper. The fact, if models produce the effects described in the publication would change the message considerably.

The structure of the publication is vey confusing. The authors mix data analysis, description of the results and conclusion in almost every section, except section 1. Some details are also provided below.

The conclusion is rather lengthy (3 pages as compared to 6 pages in the results section) and set off after stating the main results of the work folowed by further discussion of effects. This is not standard and should be changed.

I understand and accept from the analysis, that common tendencies exist. However, I wonder, if this insight is sufficient to populate a paper. Hardly ever an explanation is put forward where this correlation may come from.

Taken everything into account, I cannot recommend the manuscript for publication.

Detailed Review

Section 2: Data and methods

A discussion of the results should be left to the respective section (around line 95).

How is the ozone whole defined? By eye sight?

I have to admit: I find the equations more confusing than clarifying, especially the second one: It says: "spectral Fourier harmonics" were evaluated. But of which quantity. I suppose TOC? And how is f(\lamda) defined?

I would recommend to provide data and the software in the supplement. I doubt, that it would be possible to repeat this study using the information given.

In the whole publication I did not find, how an 'anomaly composite' is defined. I can sort of guess it, but it should be explicitly stated.

I also did not find, why sometimes NCEP and sometimes the ERA_INTERIM assimilation model is used. I would expect an argument for this also in the end Section 2, not merely stating this.

Section 3: Results Section 3.1.

Why did the authors not show the comparison for polynomials k=2 to k=6? The reader is left wondering, what exactly 'similar' means. I would suggest to include such things to the supplement, if they are not changing the message of the publication.

The authors speculate about the influence of several quantities on the TC of Ozone. Sometimes they have been observed by others, sometimes the coincidence seems obvious. However, the publication is rather descriptive. This is surely valuable in its own right. However, similar things have been described several times, as the authors long reference list shows, but often it is not clear which phenomena has been described

in which publication. An example:

The authors frequently hint vagely at some possible connection followed by a list of publications in lines 189ff. There are several statements in one sentence:

1. Significant decadal changes in the SH polar ozone are coupled with the stratospheric thermal regime

2. they may impact planetary wave propagation

3. and regional climate change in both the troposphere and the stratosphere

followed by four citations. But which publication dealt with which statement? I would expect some more guidance from a publication, than just getting a list of suggestions of what to read. So, please sort, who stated what.

Section 3.2.

The whole subsection is a listing of observed correlations, but no ordering of how important the authors think the correlation is and often without a statement of what the authors think the stated effect means.

On page 9 line 301: The authors state that: 'Close connection ... is confirmed by Fig. 8. But figure 8 shows regression coefficients. Again: correlation does not constitute causation and not even, that there is any connection.

In the end of section 3.2 (line 373 ff) I read the sentence:

'Note also that, because Figs. 6–11 present the relationships on a seasonal time scale, the statistically significant results seem to reveal new features of the troposphere–stratosphere interaction in Antarctic spring (September–November).'

Here the authors state some finding, but I dont understand what exactly they mean in the figure 6-11. Surely not all of them, because effects are sometimes cited (here in this chapter or later in the Section 4): for example: line 398-403 in the Section 4.

Discussion and Conclusion:

I think that the sentence "Our work provides further evidence that asymmetries in the distribution of Antarctic spring ozone exhibit trends and variability that relate to both tropospheric processes and the action of ozone." is not justified, because it implies that the authors describe some mechanisms of how they are related, which they did clearly not.

The authors should untangle the section 4: Discussion and Conclusion. It is very confusing to follow the authors jumps from conclusions (e.g. lines 380 till 400) to discussion i.e. line 410, which even contradicts some of the conclusions. I.e. I would understand the sentence in line 408:

'A large part of the QSW min longitude variance can be explained by the SAM-index variance (35%, Fig. 8b).'

that the SAM index variance causes the QSW min longitude variance. But in line 410 is said: 'Our results do not give information on the direction of the 'QSW min –SAM' coupling:...'

Result 3): No relationship has been demonstrated, but it has been shown, that the development is parallel even on a decadal time scale.

In line 438 ff a possible interpretation of the results is put forward. But this should clearly be discussed before stating the main results in lines 384 ff.

In the conclusions line 474 a fully new aspect comes into play: the ozone recovery. It has not been discussed before only briefly mentioned. While the statement that the recovery takes longer is backed by citation, the next sentence remains very unclear and speculative:

'... the possible influence ... the eastward shift could be renewed.'

But what does this mean in the context of the paper?

---

## Author Comment (AC1) · 11 Sep 2016

Response to Referee #1

Manuscript acp-2016-537
Evolution of the eastward shift in the quasi-stationary minimum of the Antarctic total ozone column.
A.Grytsai et al.

**RC** – Referee comments, **AC** – Author comments.
The changes, additions and corrections are in blue.

We thank Referee #1 for the helpful comments and suggestions. We have added citation and have extended discussion accordingly to **RC**. Additional citation and discussion will be included in the revised text of the manuscript.

**RC:** Page 1, L 34-35: There are earlier papers on the field and should be cited, e.g.: Chubachi, Shigeru. Preliminary result of ozone observations at Syowa station from February 1982 to January 1983. Memoirs of National Institute of Polar Research. Special issue 34 (1984): 13-19.
**AC:** We have included this paper in the citation in Chapter 1 (L 34–35). Additionally, one of the earliest publications on the Antarctic ozone decrease based on satellite data (Stolarski et al., 1986) has been also included in the citation.
L 34–35: (Chubachi, 1984; Farman et al., 1985; Chubachi and Kajiwara, 1986; Stolarski et al., 1986; Solomon, 1999).

**RC:** Page 1, L 37-38: This has already been published 20 years before Solomon et al., (2005). See for instance Chubachi, (1984) and read the fourth result in the Summary of this paper and the discussion presented in it.
**AC:** The paper Chubachi (1984) has been cited in L 38.
L 38: (Chubachi, 1984; Solomon et al., 2005).

**RC:** Page 2, L 48-53: The fundamental citations on the field of first major sudden stratospheric warming over Antarctica in Sept 2002 and the vortex split are missing.
**AC:** The three earliest papers on the event 2002 have been included in the citation.
L 46–47: The role of planetary waves was especially important in the unusual SH stratospheric warming in 2002 (Varotsos, 2002; Allen et al., 2003; Hoppel et al., 2003).
L 52–53 (wave 1 and wave 2 activity):
… September 2002 (Varotsos, 2002; Nishii and Nakamura, 2004; Newman and Nash, 2005; Peters et al., 2007; Grassi et al., 2008; Peters and Vargin, 2015).

**RC:** Page 3, L 123-124: In both formulas insert space between $m\lambda$ and $d\lambda$.
**AC:** corrected

**RC:** Page 5, L 174: Add a note that $r$ stands for the correlation coefficient and briefly discuss its statistical significance
**AC:** L 173-176: … The linear correlation between the two variables is positive and $r = 0.49–0.57$ for the seven latitude circles between 50°S and 80°S with maximum at 60°S. The correlation coefficient $r$ was calculated for the time series length $N = 35$ (1979–2013, Fig. 4) and the value $r = 0.39$ is significant at the 99% confidence limit based on Student's $t$-test. Hence, an eastward shift of the QSW minimum in the ozone distribution with high probability corresponds to a greater ozone mass deficit (larger ozone loss).

**RC:** In addition, discussion on the plausible contribution of El Nino events in 1988 and 2002 should be made. For instance, the existing literature on this subject suggests that El Nino characteristics in 1988 and 2002 may not have been similar. Brief discussion on it should be added. Furthermore, it has been suggested often that the delayed SSWs in Antarctica are being directly and primarily caused by the ozone destruction by CFCs. It would be worthwhile short discussion on this for the years 1988 and 2002 to be incorporated.

**AC:** We have added a short comment on the events 1988 and 2002.

L 177: Simultaneous negative deviations are observed in the years of large (1988) and major (2002) stratospheric warmings (vertical lines in Fig. 4). Both anomalous events in the SH stratosphere were associated with enhanced planetary wave activity (Varotsos, 2003a; Allen et al., 2003; Baldwin et al., 2003; Grytsai et al., 2008). As seen from Fig. 4, relatively small ozone mass deficit …

L 182: is opposite. The anomalies in 1988 and 2002 are the largest relatively mean tendencies in Fig. 4. Sources of anomalous planetary waves in these years have been identified in both the tropical Pacific (Kodera and Yamazaki, 1989; Grassi et al., 2008) and the SH extratropics (Nishii and Nakamura, 2004). It has been noted that the evolution of sea surface temperatures (SST) in the tropical Pacific in the spring months of 1988 and 2002 were different (Varotsos et al., 2003b) with strong La Niña and emerging El Niño conditions, respectively (see for example the monthly time series for the indices Niño 3 and Niño 4 at http://www.esrl.noaa.gov/psd/data/climateindices/). As has been identified by Lin et al. (2012), stronger stratospheric planetary wave activity is generally seen in the SH when SST anomalies exhibit La Niña–like and central-Pacific El Niño–like patterns. These authors find that a westward phase shift is seen in SH stationary planetary waves for La Niña conditions, and an eastward phase shift for warm central Pacific sea surface temperatures. Figures 3 and 5 show westward phase shift in TOC minimum and maximum longitudes in both 1988 and 2002, which in the case of 2002 is counter to the expectation from Lin et al. (2012) based on prevailing central Pacific SSTs, and which we take up further in Section 3.2. Possibly, extratropical wave sources (Nishii and Nakamura, 2004) could contribute to observed phase shift in this case. Generally, planetary wave activity contributes significantly to the interannual variability of the ozone hole size, depth and duration (Kodera and Yamazaki, 1989; Allen et al., 2003; Varotsos, 2002, 2004). On the other hand, decadal changes of the ozone hole metrics seem to be influenced by ozone depletion itself. For example, the delay in the final warming of the Antarctic vortex and seasonal disappearance of the ozone hole in 1980s–1990s appears influenced by increasing overall ozone loss (Haigh and Roscoe, 2009).

**AC:**
L 183: A general eastward shift in the TOC zonal minimum longitude in the Antarctic region occurred during 1980s–1990s.

L 287-289: However, the easternmost longitudes in Fig. 7f show also a negative anomaly in the central Pacific, as distinct from a positive one (Lin et al., 2012). Note that Fig. 7f shows also significant negative SST anomaly in the South Pacific and positive SST anomalies in the western tropical Pacific and in the Atlantic. Their combined influences could result in other phase shift direction in the SH stratosphere planetary waves than from positive anomaly in the central tropical Pacific in (Lin et al., 2012).

**RC:** Finally, comments on the discrepancies observed in the ozone vertical distribution along with TOC variability would be very informative. For example, the cases in 2001 and 2002 would be very interesting to be added.

**AC:** As the changes in the vertical ozone distribution during the austral spring influence the column ozone from year to year, we would prefer not to focus on the cases 2001 and 2002 and comment briefly these influences in terms of interannual variability.

L 344: the QSW3 pattern below the tropopause ($r_{max} \approx 0.5$) is seen. The strong correlation in the stratosphere (between 200 hPa and 20 hPa, or 12 km and 26 km, respectively, in Fig. 10) demonstrates close coupling between the $QSW_{min}$ longitude and ozone loss in their interannual variations. The vertical ozone profile in the austral spring undergoes the largest ozone decrease at this altitude range (Chubachi, 1984; Varotsos, 2003b; Solomon et al., 2005). Dynamical activity changes the extent of ozone depletion from year to year and enhances the TOC variability in the conditions of zonal asymmetry of the polar vortex. The role of zonal asymmetry becomes more important due to vertical non-alignment of the vortex structure (Varotsos, 2004). The vortex appears progressively shifted with height towards South America and high ozone in the middle stratosphere (around 30 km) within the Australian sector masks ozone depletion in the lower stratosphere (Kondragunta et al., 2005; Tully et al., 2008) resulting in additional variability of stratospheric temperature, column ozone and ozone hole characteristics. In this way, vertical changes in the wave 1 and wave 2 impacts on the ozone profile over Antarctica can contribute to the strong correlation between zonal structure in ozone distribution ($QSW_{min}$ longitude) and stratospheric temperature in Fig. 10.

**References**

Allen, D. R., Bevilacqua, R. M., Nedoluha, G. E., Randall, C. E., and Manney, G. L.: Unusual stratospheric transport and mixing during the 2002 Antarctic winter, Geophys. Res. Lett., 30, 1599, doi:10.1029/2003GL017117, 2003.

Baldwin, M., Hirooka, T., O'Neill, A., and Yoden, S.: Major stratospheric warming in the Southern Hemisphere in 2002: Dynamical aspects of the ozone hole split, SPARC Newsletter, 20, 24–27, http://www.sparc-climate.org/publications/newsletter/, 2003.

Chubachi, S.: Preliminary result of ozone observations at Syowa station from February 1982 to January 1983, Mem. Natl. Inst. Polar Res., Spec. Issue Jpn., 34, 13–19, 1984.

Grassi, B., Redaelli, G., and Visconti, G.: Tropical SST preconditioning of the SH polar vortex during winter 2002, J. Climate, 21, 5295–5303, doi: 10.1175/2008JCLI2136.1, 2008.

Grytsai, A. V., Evtushevsky, O. M., and Milinevsky, G. P.: Anomalous quasi-stationary planetary waves over the Antarctic region in 1988 and 2002, Ann. Geophys., 26, 1101–1108, doi:10.5194/angeo-26-1101-2008, 2008.

Haigh, J. D. and Roscoe H. K.: The final warming date of the Antarctic polar vortex and influences on its interannual variability, J. Climate, 22, 5809–5819, doi:10.1175/2009JCLI2865.1, 2009.

Hoppel, K., Bevilacqua, R., Allen, D., Nedoluha, G., and Randall, C.: POAM III observation of the anomalous 2002 Antarctic ozone hole, Geophys. Res. Lett., 30, 1394, doi:10.1029/2003GL016899, 2003.

Kodera, K. and Yamazaki, K.: A possible influence of sea surface temperature variation on the recent development of ozone hole, J. Meteorol. Soc. Jpn., 67(3),465–472, 1989.

Kondragunta, S., Flynn, L. E., Neuendorffer, A., Miller, A. J., Long, C., Nagatani, R., Zhou, S., Beck, T., Beach, E., McPeters, R., Stolarski, R., Bhartia, P. K., DeLand, M. T., and Huang, L.-K.: Vertical structure of the anomalous 2002 Antarctic ozone hole, J. Atmos. Sci., 52, 801–811, doi:http://dx.doi.org/10.1175/JAS-3324.1, 2005.

Nishii, K. and Nakamura, H.: Tropospheric influence on the diminished Antarctic ozone hole in September 2002, Geophys. Res. Lett., 31, L16103, doi:10.1029/2004GL019532, 2004

Stolarski, R. S., Krueger, A. J., Schoeberl M. R., McPeters, R. D., Newman, P. A., and Alpert, J. C.: Nimbus 7 satellite measurements of the springtime Antarctic ozone decrease, Nature, 322, 808–811, doi:10.1038/322808a0, 1986.

Tully, M. B., Klekociuk, A. R., Deschamps, L. L., Henderson, S. I., Krummel, P. B., Fraser, P. J., Shanklin, J. D., Downey, A. H., Gies, H. P., and Javorniczky, J.: The 2007 Antarctic ozone hole, Aust. Meteorol. Mag., 57, 279–298, 2008.

Varotsos, C.: The Southern Hemisphere ozone hole split in 2002, Environ. Sci. Pollut. Res., 9, 375–376, doi:10.1007/BF02987584, 2002.

Varotsos, C.: What is the lesson from the unprecedented event over Antarctica in 2002? Environ. Sci. Pollut. Res., 10, 80–81, doi:10.1007/BF02980093, 2003a.

Varotsos, C.: Why did a "no-ozone-hole" episode occur in Antarctica? Eos, 84, 183–185, doi:10.1029/2003EO190007, 2003b.

Varotsos, C.: The extraordinary events of the major, sudden stratospheric warming, the diminutive Antarctic ozone hole, and its split in 2002, Environ. Sci. Pollut. Res., 11, 405–411, doi:htto://dx.doi.org/10.1065/esor2004.05.205, 2004.

---

## Author Response (AR1)

acp-2016-537, 24 October 2016

Dear Editor
We appreciate Referee 2 detailed consideration of the manuscript and we revised the text according his comments and suggestions. We provided detailed response to each Referee 2 comments one by one and highlighted by blue color all changes and additions in the revised text of the manuscript. We also added one co-author (Kane Stone) and changed the sequence of authors according a value of their new contribution in modelling and discussion. These changes have been agreed with all co-authors.
On behalf of all co-authors
Gennadi Milinevsky

**Response to Referee 2**

We thank Referee #2 for the critical comments and helpful suggestions. We have revised results, discussion and conclusions accordingly to Referee Comments (**RC**). See our answers and corrections as Author Comments (**AC**).

**RC:** I have been often left wondering, what the new work in the publication is, where existing work is just continued and, most important, how relevant the work is. Mostly the authors describe phenomena and correlations. The authors frequently jump from finding a correlation of two features to the conclusion, that the connection is shown.
**AC:** We have revised and rearranged Sections 3 and 4 and have added Section 5 Conclusions and Supplement to clarify our findings and interpretation and to simplify the reading of the manuscript.

**RC:** Although the authors used models (ERA INTERIM and NCEP) they did not investigate, if the phenomena they describe are reproduced by the models. At least ERA INTERIM provides ozone also. Although they are assimilation models, it would be crucial to investigate if they reproduce the correlation described here also. The authors also failed to explain, why two models have been used. I would expect this part of the Section 2.
**AC:** We have compared two reanalysis data sets using ERA-Interim results in the main text (Figs. 6, 7, 9, 10, 11) and provide NCEP–NCAR results in Supplement for comparison (Figs. S2–S6). The results of the two reanalysis data sets agree closely.

**RC:** I understand that this is not a modelling paper and am not asking the authors to evaluate models. However, easily accessible data should have been used to help the reader to classify the paper. The fact, if models produce the effects described in the publication would change the message considerably.
**AC:** We have compared our results with published results from the ACCESS-CCM model (Stone, 2015; Stone et al., 2016) and two other studies (Lin et al., 2012; Wang et al., 2013). Discussion of the model results is included in Section 4.

**RC:** The structure of the publication is vey confusing. The authors mix data analysis, description of the results and conclusion in almost every section, except section 1. Some details are also provided below.
**AC:** As noted above, we have rearranged the manuscript to follow the Reviewer's helpful suggestions.

**RC:** The conclusion is rather lengthy (3 pages as compared to 6 pages in the results section) and set off after stating the main results of the work followed by further discussion of effects. This is not standard and should be changed.
**AC:** We have rearranged the discussion and conclusions to address this deficiency.

**RC:** I understand and accept from the analysis, that common tendencies exist. However, I wonder, if this insight is sufficient to populate a paper. Hardly ever an explanation is put forward where this correlation may come from.

**AC:** In the revised manuscript, the revealed tendencies are now based on two reanalyses and as well as new model results. We have expanded comments on the possible cause of the observed coupling in terms of troposphere–stratosphere interaction in the SH.

**Section 2: Data and methods**

**RC:** A discussion of the results should be left to the respective section (around line 95).

**AC:** We have removed the section at lines 95–99, "As seen from Fig. 1a, the TOC field in the austral spring has the two zonally asymmetric components noted in Section 1. First, the stratospheric polar vortex and ozone hole (blue) are displaced relative to the South pole and, second, the TOC field has the clear zonal maximum (red) at the subantarctic and middle latitudes. As a result, longitudinal TOC distributions demonstrate wave-1 structure in the SH midlatitude and polar zone."

**RC:** How is the ozone whole defined? By eye sight?

AC: Lines 107–110 of the revised manuscript: The thick white contour in Fig. 1a shows the ozone hole boundary, which corresponds to the threshold TOC value of 220 DU (white line on color scale in Fig. 1a) defined by the World Meteorological Organization (WMO) criterion (WMO, 2007); see also Newman et al. (2004) for the rationale for choosing the 220 DU value).

[Figure]

**Figure 1a**, modified

**RC:** I have to admit: I find the equations more confusing than clarifying, especially the second one: It says: "spectral Fourier harmonics" were evaluated. But of which quantity. I suppose TOC? And how is f(nlamda) defined?

I would recommend to provide data and the software in Supplement. I doubt, that it would be possible to repeat this study using the information given.

**AC:** Equations from Lines 110–125 have been moved to Supplement 1.

Lines 118–120 of the revised manuscript: In this work, long-term tendencies were obtained from polynomial approximation calculated with a least-squares method. The calculation method is described in Supplement 1.

RC: In the whole publication, I did not find, how an 'anomaly composite' is defined. I can sort of guess it, but it should be explicitly stated.
AC: We have been more explicit with the explanation of our anomaly composites by more clearly describing how we have evaluated climatological anomalies used in Fig 6 (Lines 209–217 of the revised manuscript), and then by describing the construction of the composites when introducing Fig 7 (Lines 264–269 of the revised manuscript).

RC: I also did not find, why sometimes NCEP and sometimes the ERA_INTERIM assimilation model is used. I would expect an argument for this also in the end Section 2, not merely stating this.
AC: We have revised Figs. 10 and 11 to present ERA-Interim results, thereby only using ERA-Interim reanalysis data in the manuscript. We have put figures using the NCEP–NCAR reanalysis in Supplement (Figs. S2–S6). We have revised the text formerly at Lines 128–131.
Lines 121–127 of the revised manuscript: Regression, correlation and composite analyses were used to relate the QSW TOC minimum (QSW$_{min}$) longitude to the meteorological variables. The reliability of the main results is examined comparing the relationships with the two reanalysis data. We use gridded atmospheric variables from the European Centre for Medium-Range Weather Forecasts (ECMWF) reanalysis ERA-Interim (Dee et al., 2011; http://www.ecmwf.int/en/research/climate-reanalysis/era-interim) at 1.5°×1.5° (longitude×latitude) resolution and NCEP–NCAR reanalysis (Kalnay et al., 1996; http://www.esrl.noaa.gov/psd/) at 2.5°×2.5° resolution.

**Section 3: Results Section 3.1.**

RC: Why did the authors not show the comparison for polynomials k=2 to k=6? The reader is left wondering, what exactly 'similar' means. I would suggest to include such things to the supplement, if they are not changing the message of the publication.
AC: Lines 140–142 of the revised manuscript: Comparison of polynomials from $k = 2$ to $k = 6$ gives similar opposite tendencies before and after the early 2000s (see Supplement 2, Fig. S1).

RC: The authors speculate about the influence of several quantities on the TC of Ozone. Sometimes they have been observed by others, sometimes the coincidence seems obvious. However, the publication is rather descriptive. This is surely valuable in its own right. However, similar things have been described several times, as the authors long reference list shows, but often it is not clear which phenomena has been described in which publication. An example:
The authors frequently hint vagely at some possible connection followed by a list of publications in lines 189ff. There are several statements in one sentence:
1. Significant decadal changes in the SH polar ozone are coupled with the stratospheric thermal regime
2. they may impact planetary wave propagation
3. and regional climate change in both the troposphere and the stratosphere
followed by four citations. But which publication dealt with which statement? I would expect some more guidance from a publication, than just getting a list of suggestions of what to read. So, please sort, who stated what.
AC: Lines 195–199 of the revised manuscript, citation has been sorted: "Significant decadal changes in the SH polar ozone are coupled with the stratospheric thermal regime (e.g., Crook et al., 2008) and, because of the zonal asymmetry in the ozone heating, they may impact planetary wave propagation (Albers and Nathan, 2012) and regional climate change in both the troposphere and the stratosphere (Gillet et al., 2009; Waugh et al., 2009)."

**Section 3.2.**

**RC:** The whole subsection is a listing of observed correlations, but no ordering of how important the authors think the correlation is and often without a statement of what the authors think the stated effect means.

**AC:** Importance of the patterns obtained from relationships in Figs. 6, 7, 9–11 is emphasized by additional comments and explanations in Lines 253–255, 286–291, 300–304, 324–326 and 394–405. We also add comments comparing the results obtained with the two reanalyses.

**RC:** On page 9 line 301: The authors state that: 'Close connection ... is confirmed by Fig. 8. But figure 8 shows regression coefficients. Again: correlation does not constitute causation and not even, that there is any connection.

**AC:** We conclude that there are multiple lines of evidence for the connection between the variables based on the statistical significance of the qualitative relationships. Although causation can not be revealed from our relationships, we try to distinguish the directions of impacts in terms of known role of dynamical disturbances in the SH atmosphere and of possible feedbacks between the stratosphere and troposphere.

On the interannual time scale, regressions and correlations show presence of SAM, QSW1 and QSW3 patterns. On decadal time scale, the patterns of QSW1 and QSW3 only demonstrate a relation to the tendency in ozone loss.

The ozone distribution passively responds to wave activity and influence of the tropospheric QSW on the ozone asymmetry (QSW1) can be reasonably assumed (Crook et al., 2008). So,

(i) presence of QSW1 in the stratosphere from year to year (Fig. 10) is caused by the upward wave propagation from the troposphere.

On decadal time scales, longitudinal shift in the QSW1/QSW3 patterns follows the tendency in ozone depletion and

(ii) feedback from the asymmetric ozone depletion on the QSW propagation (Albers and Nathan, 2012) is possible. Then, ozone depletion could contribute to long-term spatial redistribution of large-scale structures in the SH atmosphere.

As discussed in Lines 477–491 of the revised manuscript, upward and downward influences during the austral spring is possible and

(iii) on seasonal time scale, direct influence from troposphere and feedback from stratosphere can modify one another in their two-directional interaction.

We have revised Discussion dividing it into sub-Sections 4.1–4.5 to clarify our interpretation of the results.

**RC:** In the end of section 3.2 (line 373 ff) I read the sentence:
'Note also that, because Figs. 6–11 present the relationships on a seasonal time scale, the statistically significant results seem to reveal new features of the troposphere–stratosphere interaction in Antarctic spring (September–November).'
Here the authors state some finding, but I dont understand what exactly they mean in the figure 6-11. Surely not all of them, because effects are sometimes cited (here in this chapter or later in the Section 4): for example: line 398-403 in the Section 4.

**AC:** We have corrected this paragraph.

Lines 388–405 of the revised manuscript: "It can be summarized that common decadal tendencies in total ozone (Fig. 2a), QSW minimum location in the total ozone (Fig. 2b), QSW1 pattern in the lower stratosphere (Fig. 9a and 9e) and QSW3 pattern in the troposphere (Fig. 7b and 7e and Fig. 11) and lower stratosphere (Fig. 9c and 9g) exist. On interannual time scale, the SAM- and QSW1/QSW3-like patterns in the SH circulation associated with variability in the QSW minimum longitude in total ozone dominate. In general, the $QSW_{min}$ longitude changes in the TOC distribution show statistically significant associations with both the zonal mean (SAM pattern) and regional (QSW1 and QSW3

patterns) anomalies in the spring SH atmosphere. This distinguishes our results from known impacts of the spring ozone loss on the summer climate (Thompson et al., 2011). It is important to note that the revealed SAM/QSW patterns appear to be related not to ozone level itself but to the ozone anomaly location described by the the $QSW_{min}$ longitude. Possible features of the troposphere–stratosphere interaction in Antarctic spring (September–November), which could contribute to occurrence of these patterns, are discussed below."

**Discussion and Conclusion**

**RC:** I think that the sentence "Our work provides further evidence that asymmetries in the distribution of Antarctic spring ozone exhibit trends and variability that relate to both tropospheric processes and the action of ozone." is not justified, because it implies that the authors describe some mechanisms of how they are related, which they did clearly not.

**AC:** In new Section 5 Conclusions, lines 663–670, changed to: Regression, correlation, anomaly composite and model analyses show that longitudinal variability in location of the quasi-stationary zonal TOC minimum has a close relationship with variability in the TOC level itself, in the SAM/QSW patterns in the meteorological variables and in the SST patterns. Therefore, these couplings allow us to identify the SH regions, where the climate variability and climate change in austral spring could be accompanied by the ozone hole asymmetry changes. On the one hand, the results suggest combined influences of the QSW sources on the stationary wave structure in the SH stratosphere and, on the other hand, they indicate possible ozone change feedback affecting the wave structure.

In new Section 4.5 Attribution of longitude shift, possible mechanism is discussed.

**RC:** The authors should untangle the section 4: Discussion and Conclusion. It is very confusing to follow the authors jumps from conclusions (e.g. lines 380 till 400) to discussion i.e. line 410, which even contradicts some of the conclusions. I.e. I would understand the sentence in line 408:

'A large part of the QSW min longitude variance can be explained by the SAM-index variance (35%, Fig. 8b).'

that the SAM index variance causes the QSW min longitude variance. But in line 410 is said: 'Our results do not give information on the direction of the 'QSW min –SAM' coupling:...'

**AC:** Section 4 has been divided into Sections '4 Discussion' and '5 Conclusions'. Section 4 includes now sub-Sections:
4.1 Relation between the TOC asymmetry and TOC level
4.2 The SAM pattern
4.3 The QSW1/QSW3 patterns
4.4 The SST patterns
4.5 Attribution of longitude shift
As noted above, we believe that, basing on current knowledge of dynamical influence of the troposphere on stratosphere by long waves and role of stratospheric ozone depletion in tropospheric climate, reasonable assumptions on contributions of upward and downward influences to the revealed patterns could be made. Clarifying suggestions are given in the revised Discussion.

**RC:** Result 3): No relationship has been demonstrated, but it has been shown, that the development is parallel even on a decadal time scale.

**AC:** Lines 654–656 of the revised manuscript, changed to: 2) On the decadal time scale, a consistency between the longitudinal shifts of the QSW minimum in total ozone (Fig. 2b) and the QSW3 pattern in the mid-tropospheric temperature (Fig. 11) has been shown.

**RC:** In line 438 ff a possible interpretation of the results is put forward. But this should clearly be discussed before stating the main results in lines 384 ff.

**AC:** The main results have been moved to new Section 5 Conclusions, lines 647–662.

**RC:** In the conclusions line 474 a fully new aspect comes into play: the ozone recovery. It has not been discussed before only briefly mentioned. While the statement that the recovery takes longer is backed by citation, the next sentence remains very unclear and speculative:
'... the possible influence ... the eastward shift could be renewed.'
But what does this mean in the context of the paper?
**AC:** In new Section 4.5, this aspect is discussed more fully and is illustrated by the results of the ACCESS-CCM climate model (Fig. 12 and Fig. S7).

**References to Reply**

[revised manuscript text omitted]

---

## Author Response (AR3)

**Response to REFEREE # 2 report**

We thank Referee #2 for additional comments and several helpful suggestions. We include minor revisions to the text accordingly to **RCs.** We really appreciate huge work of Referee #2 that helps to improve the text of manuscript significantly.

Line numbers in **AC** are indicated for manuscript Version 4.

General:

**RC:** Compared to the first version of the manuscript the authors clarified the study and included an extensive part comparing their results to the model Australian Community Climate and Earth System Simulator (ACCESS-CCM). Still I am a bit at loss of what to make of the study. More exact, I see a list of features which all need further investigation, because many feature bring up the question how the ozone recovery will take place and how the effects on mid latitude, i.e. Australia and New Zealand will be.
**AC:** We additionally discuss regional tendencies in the two new paragraphs.
Lines 519–533:
'Note that in the years of maximum ozone hole area (easternmost $QSW_{min}$), the midlatitude wave 3 anomalies of the positive correlation partly cover New Zealand and southern tip of South America (Fig. 11c–11e, and Fig. S6c–S6e). Positive anomaly here corresponds to climate warming in the years of the easternmost $QSW_{min}$ migrations. In future, predicted ozone recovery may be accompanied by further westward shift of the wave 3 pattern and by weakening of the positive anomaly influence in region of New Zealand and South America (similarly to Fig. 11a and Fig. S6a).
As seen from Fig. 11a and Fig. S6a, both reanalyses show negative correlation anomalies over Australia and East Antarctica in the first time interval 1979–1992 (pre-ozone hole and first ozone hole years, westernmost $QSW_{min}$,). Later, these negative anomalies weaken (Fig. 11b–11d) and appear again in the latest time interval 2011–2014 (Fig. 11e). Note that regression in Fig. 6b and correlation in Fig. S2b also show negative anomaly over south-east part of Australia. These tendencies indicate that Antarctic ozone recovery to pre-ozone hole level may be accompanied by strengthening of negative coupling 'tropospheric temperature – $QSW_{min}$ longitude' in this region. All of these climate effects need further analysis.'

Additionaly, inserted in Section 5 'Conclusions':
Lines 685–686, 'Related shifts in the tropospheric QSW3 pattern play a role in climate variability in regions of Australia, New Zealand and southern tip of South America.'

**RC:** Sometimes it looks like the authors used a shotgun, fired in the bush and collected what has been hit. As I wrote in the first review, I think it is of importance and validity to hunt for features which need explaining and study. However, it should not stop here, but go on. The publication looks like a list of problems I would present to a PhD student to choose from and start investigating.
**AC:** We also agree that the analysis in our manuscript has revealed areas that require further examination, and have outlined these in the final paragraph of the conclusions.

**RC:** The authors removed several strong statements which I criticized in the first version as not being backup up by the analysis. However, I sometimes have the impression, that the authors are scarred by their own statements and back down immediately.
An example (but not the only instance): page 6 line 195ff sounds a bit trivial to me:
(less Ozone -> less heating -> weaker polar jet -> change in wave propagation) also papers are cited which find exactly this.

But the authors say "it MAY impact wave propagation" and in the next sentence: "POSSIBLE couplings ... ."

**AC:** We rephrase line 198, 'they impact planetary wave propagation' and

Line 200–201: 'Couplings between changes in the QSW structure in Antarctic total column ozone and in atmospheric variables are analyzed below.'

**RC:** In the end, in the conclusion it is finally cast in certainty. I have to admit, I find this style difficult to follow, but accept that this is a matter of taste.

In summary, I would recommend the publication of the study and hope that the authors follow their own in work in more depth and study if the connections suggested by the correlations really exist and what they mean in detail.

**AC:** In the final paragraph of the conclusions in the revised manuscript we have outlined specific areas that we regard as requiring further investigation. We hope that this information will guide further studies.

In detail:

**RC:** The authors still did not define what they mean be a 'composite'. On page 8 line 264 they write composite (average). Do they mean a composite is just an average over the three month S, O, N? Also, I still dont undestand the concept of an anomaly composit. Is it an average over several anomalies? If so, also over S, O, N?

The paragraph page 8, line 264 ff is rather confusing. It took me a while to understand this. I think one of the sentences is superfluous, because both seem to state the same thing with different words.

**AC:** Anomaly calculation procedure is described in lines 210–214: 'We first produce **monthly** climatological anomalies for each gridded **monthly average** variable at the native horizontal resolution by subtracting the associated long-term **monthly mean** (over 1979–2014 for ERA-Interim and 1981–2014 for NCEP–NCAR). We then produce **averages** of the anomalies in grid boxes of 10°×10° (latitude×longitude) **over the SON months** of each year. Finally, we evaluated…'.

Anomaly composite calculation includes anomaly averaging by criterion of extreme longitudes.

Line 267–272: 'In Fig. 7 we present anomaly composites (averages) for years of extreme western (lower $20^{th}$ percentiles) and eastern (upper $20^{th}$ percentiles) $QSW_{min}$ longitudes to further investigate the patterns shown in Fig 6. Monthly mean anomalies for September, October and November were calculated by subtraction of the climatological means of 1979–2014 from monthly mean variable value in each grid box as described above concerning Fig. 6. Then monthly mean anomalies were averaged over the SON months.'
'

**RC:** page 6 line 177 Do the authors mean, that r=0.39 would already be significant?

**AC:** Lines 178–179, corrected: '…and the value $r = 0.43$ is significant at the 99% confidence limit based on a two-tailed Student's $t$-test.'

**RC:** page 7 line 212/13 Wording: Two sentences start with 'We then ... '

**AC:** Line 214: 'We then evaluated the regression…' changed to

'Finally we evaluated the regression…'

**RC:** A sentence like line 252 ...variability ... with high probability ... could ... does not sound as if the authors are convinced by there own study. It is another example of what I wrote before, that the authors sometimes seem afraid of their own findings.

**AC:** Lines 254–256: 'Hence, variability in zonal asymmetry in the Antarctic ozone during the spring months, with high probability, is indicative of the SH regional climate variability.'

**RC:** line 256 - 259 Is this in contradiction to (Mo and Higgins, 1998)?
**AC:** The Pacific wave train is present in many relationships (Fig. 9f, Fig. 11, Fig. S2c, Fig. S3d and Fig. S4d), in consistency with (Mo and Higgins, 1998). Our important result is that combined wave activity over the three ocean basins can contribute to the $QSW_{min}$ longitude variability as noted in lines 325–328.

We somewhat modify lines 258–262:
'Zonal asymmetry in the SH troposphere circulation is closely coupled with the Pacific–South American (PSA) mode (Mo and Higgins, 1998). The PSA pattern in the RC distribution in Fig. 6 is of insignificant intensity, whereas pronounced meridional wave trains are seen in Indian–Australian sector and Atlantic–South American sector (U200 in Fig. 6c). As follows from the relationships below, combined wave activity over the three ocean basins can contribute to the $QSW_{min}$ longitude variability.'

**RC:** page 9 line 313 Do the authors mean, the the surface pressure anomalies cause the variance of the OSWmin longitudinal variance?
**AC**: Since the QSW3 and SAM activities in the SH tropospheric circulation influence wave penetration into the stratosphere, the surface pressure anomalies cotribute to the $OSW_{min}$ variations. However, we note also that feedback processes are possible (lines 424–427, 446–450, 457–458).

Lines 315–317, corrected: 'Therefore, approximately 57% and 35% of the QSWmin longitude variance (both significant at the 95% confidence limit) can be explained by the surface pressure anomaly variance described by regional (grid box 1, QSW3 pattern) and hemispheric-scale (SAM) indices, respectively.'

**RC:** page 10 line 322 ... show likely ... in connection with line 325
... could mean ... makes a very weak statement: An assumption which may be true leads probably to a cause.
**AC:** Line 325, The U200 anomaly composites in Fig. 9b and 9f show meridional …
Line 327–328, Relation of the QSWmin longitude to these wave patterns means combined contribution...

**AC:** We add also keywords.
Lines 35–36:

[revised manuscript text omitted]

**Supplement 1.** Calculations of the long-term tendencies for timeseries presented in Fig. 2. Polynomial approximation was calculated with a least-squares method by a following scheme (see, e.g. Krishnan, V.: Probability and random processes, John Wiley & Sons, Inc., Hoboken, New Jersey, 420 p., 2006). We consider $n$ pairs of values $(x_i, y_i)$. In our calculations $x_i$ and $y_i$ are time and the phase of the TOC distribution (e.g. in the quasi-stationary minimum at 65°S), respectively. The purpose is to find a polynomial fit of degree $k$ to minimize

$$f(x_i, y_i) = \sum_{i=1}^{n} (P_k(x_i) - y_i)^2$$

where

$$P_k(x_i) = \sum_{m=0}^{k} a_m x_i^m$$

is a polynomial of $k$-th power with unknown coefficients $a_j$. The minimization condition for the function $f$ is written as

$$\frac{\partial f}{\partial a_j} = 0,$$

which includes a system from $k + 1$ equations for different $j$. In solving the system we obtain each coefficient as the ratio of the two determinants:

$$a_m = \frac{\begin{vmatrix} n & \sum_{i=1}^{n} x_i & \dots & \sum_{i=1}^{n} x_i^{m-1} & \sum_{i=1}^{n} y_i & \dots & \sum_{i=1}^{n} x_i^k \\ \dots & \dots & \dots & \dots & \dots & \dots & \dots \\ \sum_{i=1}^{n} x_i^k & \sum_{i=1}^{n} x_i^{k+1} & \dots & \sum_{i=1}^{n} x_i^{k+m-1} & \sum_{i=1}^{n} x_i^k y_i & \dots & \sum_{i=1}^{n} x_i^{2k} \end{vmatrix}}{\begin{vmatrix} n & \sum_{i=1}^{n} x_i & \dots & \sum_{i=1}^{n} x_i^{m-1} & \sum_{i=1}^{n} x_i^m & \dots & \sum_{i=1}^{n} x_i^k \\ \dots & \dots & \dots & \dots & \dots & \dots & \dots \\ \sum_{i=1}^{n} x_i^k & \sum_{i=1}^{n} x_i^{k+1} & \dots & \sum_{i=1}^{n} x_i^{k+m-1} & \sum_{i=1}^{n} x_i^{k+m} & \dots & \sum_{i=1}^{n} x_i^{2k} \end{vmatrix}}$$

**Supplement 2.** Polynomial fits for the timeseries of the QSW minimum longitude at 65°S in comparison with Fig. 2b ($k = 3$).

[Figure]

**Figure S1:** Longitude of the QSW minimum at 65°S averaged for September–November. Thin lines are timeseries of 1979–2015 and thick lines are polynomial fits of degree $k = 2, 4–6$ .

**Supplement 3.** Images in Figs. S2–S6 based on data by the NOAA/ESRL Physical Sciences Division, Boulder Colorado from their Web site at http://www.esrl.noaa.gov/psd/.

[Figure]

**Figure S2:** Correlations between the TOC QSW minimum longitude against NCEP–NCAR reanalysis climatological anomalies of (a) surface pressure (SP), (b) surface temperature, (c) 200-hPa zonal wind speed (U200) and (d) 200-hPa vertical pressure wind speed (W200) for SON 1979–2014. Black (white) contours show positive (negative) correlations significant at the 95% confidence limit. To compare with Fig. 6 based on the the ERA-Interim data.

[Figure]

**Figure S3:** As in Fig. 7, but for the NCEP–NCAR reanalysis.

[Figure]

**Figure S4:** As in Fig. 9a, 9b, 9e and 9f, but for the NCEP–NCAR reanalysis.

Figs S5 and S6 show correlations between the QSWmin longitude and NCEP–NCAR air temperature to compare with those from the ERA-Interim data presented in Fig. 10 and Fig. 11, respectively.

[Figure]

**Figure S5:** Longitude–height cross-section of the correlation between the QSW$_{min}$ longitude at 65°S and air temperature averaged over the zone 40–60°S for SON 1979–2014. Thick black curve marks climatological thermal tropopause from the NCEP–NCAR reanalysis. Black (white) contours show positive (negative) correlations significant at the 95% confidence limit.

[Figure]

**Figure S6:** Correlation between the QSW$_{min}$ longitude at 65°S and air temperature at 500 hPa south of 30°N. Five sequential 14-year intervals with 5–6 year step are presented. Black (white) contours show positive (negative) correlations significant at the 95% confidence limit. Thick solid black (dashed white) lines mark mean longitudinal positions of the positive (negative) correlation peaks in the QSW3 structure.

Fig. S7 shows the meridional cross-section of the linear correlation between temperature and QSW$_{min}$ longitude from the ACCESS-CCM REF-C2 and REF-C1 simulations.

[Figure]

**Figure S7:** Similar to Fig. 10, but showing the longitude–height cross-section of the correlation between the QSW$_{min}$ longitude at 65°S and air temperature averaged over the zone 40–60°S for (a) the REF-C2 simulation over 1979–2014 and (b) the REF-C1 simulation over 1979–2010.